# A model of mercury cycling and isotopic fractionation in the ocean

David E. Archer[1], Joel D. Blum[2]

[1]Department of the Geophysical Sciences, University of Chicago, Chicago, 60637, USA
[2]Department of Earth and Environmental Sciences, University of Michigan, Ann Arbor, Michigan 48109

*Correspondence to*: David E. Archer (d-archer@uchicago.edu)

**Abstract.** Mercury speciation and isotopic fractionation processes have been incorporated into the HAMOCC offline ocean tracer advection code.  The model is fast enough to allow a wide exploration of the sensitivity of the Hg cycle in the oceans, and of factors controlling human exposure to monomethyl-Hg through the consumption of fish.  Vertical particle transport of

Hg appears to play a discernable role in setting present-day Hg distributions, which we surmise by the fact that in simulations without particle transport, the high present-day Hg deposition rate leads to an Hg maximum at the sea surface, rather than a subsurface maximum as observed.  Hg particle transport has a relatively small impact on anthropogenic Hg uptake, but it sequesters Hg deeper in the water column, so that excess Hg is retained in the model ocean for a longer period of time after anthropogenic Hg deposition is stopped.  Among 10 rate constants in the model, steady state Hg concentrations

are most sensitive to reactions that are sources or sinks of Hg(0), the evasion of which to the atmosphere is the dominant sink term in the surface ocean. Isotopic fractionations in the interconversion reactions are most strongly expressed, in the isotopic signatures of dissolved Hg, in reactions that involve the dominant dissolved species, Hg(II), including mass independent fractionation during Hg photoreduction.  The $\Delta^{199}$Hg of MMHg in the model, subject to photoreduction fractionation, reproduces the $\Delta^{199}$Hg of fish in the upper 1000 m of the ocean, while the impact of anthropogenic Hg deposition on Hg

isotope ratios is essentially negligible.

## 1 Background

The element mercury (Hg) is a powerful neurotoxin(Clarkson and Magos, 2006).  When transformed to methyl mercury (MeHg) it is known to amplify its toxicity by bio-accumulating up the food chain. The main human exposure to MeHg is via consumption of high trophic level seafood (Chen et al., 2016;Schartup et al., 2018).  Humans have been mining and

mobilizing Hg into the Earth surface environment for hundreds of years, as a by-product of coal combustion, and for its use in gold mining, and in products such as electronics and light bulbs (Amos et al., 2013;Driscoll et al., 2013;Krabbenhoft and Sunderland, 2013;Lamborg et al., 2014;Obrist et al., 2018;Streets et al., 2017;Mason et al., 2012).  The Hg load in the surface ocean has increased by a factor of 3-5 since the industrial revolution; this represents a massive human impact on the global Hg cycle (Streets et al., 2017).

Hg can be extremely mobile in the environment, with gaseous forms in the atmosphere, and with particle-reactive forms allowing it to travel through soils and rivers and into the oceans (Fitzgerald et al., 2007). Hg(II) has a high affinity for complexing with (or adsorbing to) sulfur-rich ligands in organic matter (Schartup et al., 2015) and this leads to Hg accumulation with organic carbon in soils (Amos et al., 2013;Smith-Downey et al., 2010;Biswas et al., 2008) and sediments (Hollweg et al., 2010). The high mobility of Hg implies that the amount of Hg in earth surface reservoirs is transient, even in the steady-state pre-human Hg cycle (Amos et al., 2013), and that Hg can be potentially mobilized by human impacts such as the thawing of Arctic permafrost (Schuster et al., 2018;Obrist et al., 2017) or enhanced wildfire activity (Turetsky et al GRL 2006).

The Hg cycle is analogous to the carbon cycle, in which fossil fuel extracted from the solid Earth is released to a fast surface system consisting of soils and oceans in communication via the atmosphere.  In both cases, the long-term sink for the perturbation is burial in sediments of the ocean.  Because these burial fluxes are relatively slow, it will take a long time for these perturbations to subside: thousands of years for the Hg cycle (Amos et al., 2013), and hundreds of thousands of years for the carbon cycle (Archer et al., 2009).  Other forms of environmental degradation that will persist for thousands of years include actinide radioactive waste, and some anthropogenic gases such as sulfur hexafluoride (Ray et al., 2017).

It is extremely challenging to predict the future of human exposure to Hg, because the Hg cycle is so complex (Blum, 2013). One challenge has been to characterize the quantitative role of Hg adsorbed onto sinking particles in the ocean (Lamborg et al., 2016), which will constrain how deeply anthropogenic Hg may have penetrated into the ocean (Lamborg et al., 2014;Munson et al., 2015;Zhang et al., 2014b).  Another is to understand the factors that control the production of MeHg, which is the bio-accumulating form but which comprises only a small fraction of the Hg in the ocean (Schartup et al., 2015;Schartup et al., 2013;Ortiz et al., 2015;Lehnherr et al., 2011;Lehnherr, 2014;Jonsson et al., 2016;Chakraborty et al., 2016;Blum et al., 2013).

Stable isotopes provide a powerful tool for determining the origins (Kwon et al., 2014;Li et al., 2014;Sherman et al., 2015;Sherman et al., 2013;Balogh et al., 2015;Demers et al., 2015;Donovan et al., 2014;Donovan et al., 2013;Gehrke et al., 2011;Sun et al., 2016;Tsui et al., 2014;Sonke JE, 2010;Yin R, 2013) and transformations (Kwon et al., 2013;Kwon et al., 2014;Rodriguez-Gonzalez et al., 2009;Chandan et al., 2015;Yang and Sturgeon, 2009;Foucher D, 2013;Jiskra M, 2012),  of Hg in the natural environment.  Hg has seven stable isotopes with six at high abundance (>1%).  Most chemical processes fractionate the various isotopes progressively according to their masses (Mass Dependent Fractionation; MDF).  If all fractionation processes were strictly mass dependent, measurements of the proportions of more than two isotopes would be redundant information.  However, Hg is susceptible to light-stimulated reactions, which include oxidation of Hg(0) and reduction of Hg(II) and MeHg. These photochemical reactions exhibit MDF and Mass Independent Fractionation (MIF), which distinguishes between isotopes beyond their mass differences (Blum et al., 2014;Bergquist and Blum, 2009).  Odd mass number mass independent fractionations, or (odd-MIF), are produced by two mechanisms. Large magnitude effects (>~0.4‰) are seen in kinetic short-lived radical pair reactions and are believed to be caused by the magnetic isotope effect (Buchachenko, 2001;Bergquist and Blum, 2009). Smaller magnitude odd-MIF can also be produced during dark equilibrium

reduction and oxidation reactions by the nuclear volume effect (Schauble, 2007;Zheng and Hintelmann, 2010). Even-MIF has been observed in Hg in the atmosphere (Gratz et al., 2010) and deposited from atmospheric sources (Strok et al., 2015;Zheng et al., 2016), and is believed to occur in the tropopause, but the specific mechanism is not known (Chen et al., 2012). Mass independent fractionation provides multiple degrees of freedom, allowing measurements of the proportions of
all the isotopes to carry much more information than would be possible if only MDF occurred.

We have incorporated a model of the chemical transformations and isotopic fractionations of Hg in the ocean into the HAMOCC offline ocean passive tracer advection model (Maier-Reimer and Hasselmann, 1987). The flow field is taken from the Large Scale Geostrophic (LSG) dynamics model, which is also extremely fast and efficient for 3-D ocean flow (Maier-Reimer, 1993). The LSG physical model takes a time step of a month by eliminating non-geostrophic parts of the
circulation that would be violated by this extremely long time step. The HAMOCC tracer advection model takes an annual average flow field from 12 monthly time steps of the LSG model and uses it to advect tracers through the ocean. While the tracers are flowing, they exchange with the atmospheric gases (in the case of $CO_2$ and $O_2$), and with biota (as $CO_2$, $O_2$, alkalinity, and nutrients).

The distribution of Hg in the ocean today is the product of a presumably steady state natural Hg cycle, which takes thousands
of years to achieve in the model due to the ocean turnover time, followed by a global human perturbation which began in about 1850 (and which could persist for thousands of years into the future). HAMOCC is believed to still be the fastest off-line 3-D ocean tracer advection code in existence and is ideal for studying the sensitivity of the ocean Hg cycle on these long time scales. This paper is also the first attempt in our knowledge to simulate the isotopic fractionation processes of Hg in the ocean, which take thousands of years to express themselves globally.

**2 Modeling Methods**

**2.1 Mercury Geochemistry Solvers**

The geochemical cycling of Hg in the ocean in Hg-HAMOCC is similar in conception to previous models (Figure 1). Hg interconverts between Hg(II), Hg(0), monomethyl-Hg (MMHg), dimethyl-Hg (DMHg), and Hg adsorbed to sinking particles (Hg-P). The rates of the biological reactions are correlated to each other and to the overall rate of metabolic activity in the
(Semeniuk and Dastoor, 2017) model and in our model, with typical values as shown in Figure 1. The rate constant for MMHg production from Hg(II) is proportional to the rate of POC degradation, which is derived from the attenuation with depth of the sinking POC flux in HAMOCC (expressed as a volumetric rate of POC degradation).

Other Hg transformation reactions are provoked by light (Blum et al., 2014;Bergquist and Blum, 2007), and only take place in the surface ocean. The rate of photochemical reactions in Hg-HAMOCC is about a factor of two higher in low versus
high latitudes, using the latitudinal function that governs export production rates in HAMOCC. The photochemical reaction rates are attenuated with water depth, using an e-folding depth scale of 20 meters. The wavelength dependence of photochemical reactions and fractionations is complex (Rose et al., 2015), and the attenuation depth of the light varies with

frequency, so the 20 m depth scale is only an approximation. The actual mechanisms for DMHg production and degradation are still uncertain, so the model formulation can be regarded as something of placeholder for the time being. The rates of gas evasion of Hg(0) and DMHg are taken to be proportional to the concentrations of the species. The Hg cycle in the surface ocean is driven by deposition influx of Hg(II) and gas invasion of Hg(0), which are applied at uniform rates around the world.

All of the rate constants in Hg-HAMOCC are first-order, which is to say that the chemical rates are determined by multiplying the rate constant by a single species concentration to the first power. Rates of conversion between these species are generally fast, some much faster than the 1-year time step of the tracer code. For this reason, solvers were written to find steady state distributions of the Hg species. Because the Hg system is strongly driven at the sea surface by air-sea fluxes, a separate solver system was developed for surface grid points than the one applied to subsurface grid points.

Export of sinking Hg-P is done separately from the speciation calculations in subsurface waters, but simultaneously with the speciation calculations in the surface ocean. Hg advection by ocean circulation is also done in an independent step from the chemistry and particle components. Since the Hg speciation is imposed to be at equilibrium by the speciation solvers, there is no need to carry speciation information through the advective system, which only needs to carry around a single tracer for the total Hg concentration. To treat the isotopic systematics of the Hg cycle, we added three additional advected Hg tracers, identical to the first but with slightly altered source fluxes or rate constants, in order to simulate variations in the relative abundances of isotopes [199]Hg, [200]Hg, and [202]Hg relative to [198]Hg.

### 2.1.1 Surface ocean chemistry solver

For the surface ocean, the distribution of Hg among the dissolved species is determined by a balance of Hg fluxes through the system: rain input of Hg(II) and Hg(0), and removal by Hg(II) scavenging on sinking particles and degassing as Hg(0) and DMHg. Concentration-dependent reaction rates in the model are all assumed to be first-order, i.e. linear in Hg concentration. This includes loss by gas evasion, which should be linear in Hg concentration in the piston velocity model, and loss of bound Hg on sinking particles, which is linear with [Hg(II)] in the adsorption model. The solver finds values for the Hg species concentrations at which the incoming and loss fluxes balance. The equations are:

$$
\begin{bmatrix}
-k_{20} - k_{2M} - k_{2D} - S & k_{02} & k_{M2} & 0 \\
k_{20} + k & -k_{evp} - k_{02} & k_{M0} & 0 \\
k_{2M} & 0 & -k_{M2} - k_{M0} - k_{MD} & k_{DM} \\
k_{2D} & 0 & k_{MD} & -k_{evp} - k_{DM}
\end{bmatrix}
\begin{bmatrix}
[Hg(2+)] \\
[Hg(0)] \\
[MMHg] \\
[DMHg]
\end{bmatrix}
=
\begin{bmatrix}
-Dep_{Hg(2+)} \\
0 \\
\\
-Dep_{Hg(0)}
\end{bmatrix}
$$

where k denotes a first-order rate constant, subscripts denote reactant then product where 2 = Hg(II), 0 = Hg(0), D = DMHg, and M = MMHg. The Dep terms on the right-hand side denote deposition from the atmosphere at fixed imposed rates. S is a rate constant for Hg(II) sinking on particles

$$S = \left(1 - \frac{1}{k_b[POC]+1}\right)\frac{R}{dz}$$

comprised of the POC concentration, the scavenging constant, and an imposed POC sinking velocity R. The diagonal terms in the first matrix represent sinks of the chemical species listed in the second matrix, when those sinks are calculated as the rate constants in the diagonal multiplied by the species concentration (which is solved for). The positive off-diagonal terms in the first matrix represent sources of species, which are calculated as rate constants times the concentration of the origin species. This linear algebraic calculation solves for the steady-state concentrations of the four Hg species without iteration (in contrast to an analogous solver in HAMOCC for $CO_2$ system chemistry).

### 2.1.2 Subsurface ocean chemistry solver

The Hg cycle in the deep ocean differs from that of the surface in that fluxes of Hg into and out of the system (by desorption of Hg(II) from particles) are slow relative to the rates of interconversion between the Hg species. Because all of the rate constants are first-order, the relative proportions of the species are independent of the total Hg concentration. The solver finds steady state values of all species relative to that of Hg(II), then scales everything to fit the total Hg concentration as produced by the advection routine. The equations are:

$$\begin{bmatrix} -k_{02} & k_{M2} & 0 \\ 0 & -k_{M0}-k_{M2}-k_{MD} & k_{DM} \\ 0 & k_{MD} & -k_{DM} \end{bmatrix} \begin{bmatrix} [Hg(0)] \\ [MMHg] \\ [DMHg] \end{bmatrix} = \begin{bmatrix} -1pM \times k_{20} \\ -1pM \times k_{2M} \\ -1pM \times k_{2D} \end{bmatrix}$$

where a concentration of 1pM is assumed for [Hg(II)], in order to work out the relative proportions of the other species. After the proportions of all the species concentrations are worked out, they are scaled to match the total Hg concentration as it is slowly changed by advection and desorption from POC. Maps of Hg total concentration are compared with measurements in Figure 2, and profiles of the Hg species are shown in Figure 3. In addition to the global mean, profiles from the highly productive equatorial Pacific to the oligotrophic Atlantic are shown to span the range of variability in the model.

### 2.2 Hg adsorption and transport on particles

Hg has a strong chemical affinity for organic matter, in particular for organic sulfur ligands. This chemistry leads Hg to adsorb onto organic matter in the ocean, leading to a vertical sinking flux of adsorbed Hg on particles (Lamborg et al., 2016). Characterizing this flux is complicated by the fact that sinking particles compete for Hg with suspended and dissolved organic carbon (Han et al., 2006;Fitzgerald et al., 2007).

The biological pump in HAMOCC is represented as an instantaneous vertical redistribution of nutrients and other associated biological elements, without ever resolving them into particles or tracking their sinking. We constructed a hypothetical POC profile from this functioning of HAMOCC by choosing a POC sinking velocity that would transform the export production from the euphotic zone in HAMOCC into surface POC concentrations that are close to the observed mean concentration of about 5 μM. This sinking velocity of 500 meters per year is much slower than the 100 meters per day inferred sinking velocities of the particles that carry the bulk of the material caught in sediment traps, but it is similar to that used by other recent estimates (Semeniuk and Dastoor, 2017;Lamborg et al., 2016), and comparable to the result of modeling thorium on particles (Anderson et al., 2016), which (similarly to Hg) binds to both suspended and sinking particles. Because POC in the real ocean spans in size from dissolved to fast-sinking, the imposition of a single velocity in the model formulation, to be applied to the entire adsorbed Hg pool, is an oversimplification of reality, and the velocity required for the best fit is not a simple thing that can be measured directly in the real ocean. The sensitivity of the model to the sinking velocity is shown in Figure 4. With an increase in sinking velocity, the "biological pump" for Hg becomes stronger, increasing the concentration in the deep ocean. The scavenging lifetime of Hg decreases as the sinking flux increases with increasing sinking velocity. When the Hg sinking velocity is set to 500 m/yr, the same velocity as is used to transform the POC flux into a POC concentration, the global Hg sinking rate is similar to the result of Semeniuk and Dastoor (2017) (Figure 5). However, sinking fluxes of Hg in the mid water column are about a factor of three lower than equatorial and North Pacific sediment trap fluxes from Munson et al. (2015), so the models could be under-predicting the real particle fluxes if the Munson et al. (2015) data are globally representative. The turnover time of dissolved Hg, with respect to transiting through the water column on sinking particles, depends on the sinking velocity, as shown in Figure 4. Values approaching 1000 years in the deep ocean have been reported in other models (Semeniuk and Dastoor, 2017;Zhang et al., 2014a) and imply that circulation plays a major role in determining deep ocean Hg concentrations.

A second degree of freedom in the system of sinking Hg on particles is the adsorption constant $K_d$, defined such that:

$$[Hg\text{-}P] / [Hg(II)] = K_d [POC] \hspace{3cm} \text{(eqn. 1)}$$

Semeniuk and Dastoor (2017) and Zhang et al. (Zhang et al., 2014b) used a value of $2\cdot10^5$, in units of L kg$^{-1}$ (requiring POC to be in kg L$^{-1}$), but included a factor of 10 correction for the fraction of particulate material in the ocean that is organic carbon ("$f_{oc}$"), resulting in an effective $K_d$ of $2\cdot10^6$. Lamborg et al. (2016) derived a value of about $4\cdot10^6$, which they claim to be a factor of 20 higher than the value used in the models, but after the $f_{oc}$ correction in the models the values only differ by a factor of 2. The data from Bowman et al. (2015), analysed by Lamborg et al. (2016), showed that about 5% of the Hg(II) in surface waters is bound to sinking particles, similar to the results from the models (Semeniuk and Dastoor, 2017). A map of the bound fraction in surface waters from our model is shown in Figure 6, showing more particulate Hg in high-production (high POC) regions such as the equatorial Pacific. The sensitivity of the model to the value of $K_d$ is shown in Figure 4, with results similar to those for sinking velocity. The calculated lifetime of dissolved Hg in the water column, relative to removal by adsorption to sinking particles, is shown in Figure 7. The figure is intended for comparison with results of other models,

and to show that the Hg cycle in the ocean is close to a crossover point between dominance by fluid advection vs. sinking particles.

## 2.3 Isotopic fractionation

Mercury isotope fractionations associated with any of the processes in the Hg cycle are treated in the model as kinetic effects; slight perturbations in the rates of chemical transformations between the isotopes (rather than fractionation of the equilibrium state). This allows Hg-HAMOCC to impose fractionation effects onto the kinetic expressions in the solvers for surface and subsurface Hg speciation. The altered kinetic rate constants are applied to alternative total Hg concentration fields representing the isotopes $^{199}$Hg, $^{200}$Hg, and $^{202}$Hg relative to a "base" isotope $^{198}$Hg. For many processes, such as advection by fluid flow and mixing, isotopic delta values can be manipulated directly as a tracer. However, to calculate the expression of an isotope fractionation in the steady state solution to a web of chemical reactions requires that we simulate the impact of the fractionation on the budgets of the different isotopes, as the simplest way to come up with a delta value.

The way that the model treats the isotopes differs from reality, for a numerical convenience, following a technique developed by Ernst Maier-Reimer in HAMOCC many years ago for carbon isotope ratios (Maier-Reimer, 1984). In the real world, the total Hg concentration is comprised of multiple isotopes. In the model, the concentrations of Hg in the ocean are taken as that of a base isotope. Then the entire Hg cycle in the model is duplicated, and the kinetic constants slightly altered, to represent the behaviour of a different Hg isotope. Each isotopic field corresponds to how the total Hg field would behave if it were entirely comprised of its particular Hg isotope, subject to slightly altered sources and kinetic rate constants for that isotope.

The deviations of the fields for the other isotopes are represented as ratios relative to the base field, and presented in permil (‰), where the ratio of the "standard" is 1 rather than the particular ratio of the isotopic reference standard for natural samples. The relative differences, represented as ratios in ‰, are the same between variations in the isotopes in reality and between the altered fields in the model, even though the concentrations are different between the two cases. The advantage of this scheme is that the fields representing the different isotopes are subjected to similar computational rounding errors, because their values are similar. Also it is simpler to simulate the behaviour of total Hg in a single field, rather than as a more complicated sum of isotopic concentrations as in reality.

Because the chemical speciation of Hg is solved for each time step, there is no need to advect the concentrations of chemical species such as MMHg. The advection scheme in the model carries the total concentrations representing each isotope. Each isotope field is divided into the different Hg species assuming steady state and using the web of kinetic rate constants appropriate to that isotope. The slightly altered speciation of one isotope relative to another, and the slightly differing sources and sinks for that isotope, lead to slight differences between the abundances of each isotope overall in the Hg pool.

Mass-dependent fractionation processes are imposed on all isotopic systems, with the rates depending on how much heavier an isotope is than mass 198. Mass-independent fractionations in the ocean are applied only to the $^{199}$Hg system, while the $^{200}$Hg system is driven only by different isotopic signatures of wet (Hg(II)) vs. dry (Hg(0)) deposition. Mass independent

fractionations are calculated by subtracting the expected mass dependent fractionation, to produce a composite quantity Δ value. The solver finds the impact of the fractionation mechanisms on the steady-state isotopic signatures in the Hg system: the expression of the isotope effects within the kinetic ocean Hg cycle.

## 2.4 The Anthropogenic Perturbation

Human activity has resulted in significantly increased Hg emission to the global biosphere since about 1850 (Streets et al., 2011;Streets et al., 2017;Amos et al., 2013;Horowitz et al., 2014), which has lead to an increase in Hg deposition to the ocean. Because of the tendency for Hg to recycle in the environment, the relationship between emissions and deposition is not simple and immediate, but rather reflects the entire cumulative emission and re-emission of Hg. Guided by a reconstructed history of atmospheric Hg through time (Streets et al., 2017), we subject our model to a 4 times increase in Hg

deposition, following an initial spin-up equilibration period of 10,000 years. The beginning of the anthropogenic period corresponds to approximately the year 1850. We show natural steady-state results from model year 1850, which are useful for understanding how the ocean Hg cycle works, and contemporary results from model year 2010, for comparison with field measurements. Anthropogenically enhanced deposition is continued at a constant rate until the year 2100, after which we follow two scenarios: an abrupt and unrealistic return to natural Hg deposition fluxes, useful to determine the time constant

of the oceanic recovery, and a "hangover" scenario in which an abrupt cessation of human Hg emissions triggers a gradual slowdown of enhanced deposition, over an ocean overturning time scale of 1,000 years.

## 2.5 Method Limitations

The steady state assumption in the Hg solvers limits the ability of Hg-HAMOCC to explore detailed shallow-water interactions of turbulence, ventilation, and photochemistry, and the physics of the tracer advection code preclude exploration

of processes on short time scales, such as the seasonal cycle near the surface. The model allows us to explore the interaction of the Hg chemistry and particle adsorption with the ocean circulation on long time scales.

A peculiarity of the surface ocean solver is that fluxes of Hg across the sea surface are always locally balanced, by construction, neglecting the impact of any upwelling Hg driving sea surface Hg concentrations and evasion rates to higher values. Similarly to the treatment of $O_2$ gas in HAMOCC (Maier-Reimer and Bacastow, 1990), the Hg concentrations in the

surface box (50 m) are maintained at atmospheric saturation through the iterations in the advection scheme. The concentrations in the box below that (to 125 m) are comprised of 25% saturation while the other 75% is driven by subsurface advection. Because Hg concentrations in the top box are determined by a balance of fluxes with the atmosphere, in places where surface divergence brings up Hg from below, the advective upwelling source is missed by Hg-HAMOCC, which will underestimate the Hg surface concentrations and degassing rates somewhat. To use the model to simulate a transient uptake

of Hg by the ocean in response to a change in the surface rain rate, we can track the change in global ocean inventory of Hg with time, but the fluxes determined by the solver at the air-sea interface will balance to zero, locally and at all times, defiant

of the net fluxes that are filling the deep ocean with Hg. The top box of the model (50 m) serves as a sort of boundary condition for Hg.

## 3 Results

### 3.1 Particle Sinking versus the Overturning Circulation

There are two competing mechanisms for Hg invasion into the deep ocean: advection by the overturning circulation and the flux of Hg adsorbed on sinking particles. We use our model to explore the interaction of these pathways. There are two end-member cases to consider; one with particles dominating the distribution and transport of Hg, and the other with circulation dominating. The particle-flux dominating end member condition can be achieved in Hg-HAMOCC by disabling the advection of the Hg tracers (Figure 8, orange line). In the steady state, in order to achieve Hg concentrations that are not

changing through time, the vertical flux of Hg through the water column must be the same at all depth levels. The flux of sinking POC decreases with depth in the ocean due degradation. The abundant POC sinking flux in the surface ocean carries the same Hg sinking flux as the rarefied POC sinking flux in the deep sea.

This means that in the steady state, the POC in the deep sea has to carry more Hg than it would in the surface ocean. The adsorbed Hg is linearly related to the dissolved Hg by the adsorption equation (1). Rearranging (1) gives:

[Hg-P] = [Hg(II)] Kd [POC]                                    (eqn. 2)

If we take the sinking Hg-P flux to be proportional to [Hg-P] (assuming a uniform sinking velocity), then a decrease in the flux of POC (proportional to [POC] for the same reason), requires a higher dissolved [Hg(II)]. The result is that, in the steady state, Hg concentrations rise with depth in the ocean, to compensate for the decrease in sinking POC flux. A smaller POC sinking flux will have to carry a higher Hg concentration in order to sustain the required depth-uniform Hg flux, and

the higher adsorbed Hg concentration requires a higher Hg concentration in the water column.

The other end-member case comes much closer to the observed distribution of Hg in the deep ocean. When circulation dominates, and particle transport of Hg is disabled, the Hg concentrations maintained in the surface ocean (by balancing evasion against deposition) are imposed on the deep ocean, resulting in a nearly uniform distribution of Hg throughout the ocean (Figure 8, blue line). There are some regional variations in Hg in this scenario, but they are not systematic, as

compared to the clear Pacific–Atlantic differences exhibited by nutrient-type elements (concentrated in the Pacific) versus by strongly scavenged elements like Al (concentrated in the Atlantic, where deposition is more intense).

The balance between advection versus sinking particles affects the uptake of anthropogenic Hg by the ocean. Profiles of total Hg changes from pre-anthropogenic to present-day, after 130 years of enhanced Hg deposition (to 2010), are shown in Figure 3. If particles are neglected or sink so slowly as to be negligible in the Hg cycle, there is a sharp surface spike in Hg

concentrations in the model simulation of the present-day (2010), due to increased deposition. An increasing importance of particle transport tends to moderate a surface ocean spike, while transferring much of the anthropogenic Hg load to a

subsurface maximum corresponding to the location of POC degradation in the thermocline. Particulate Hg transport to depth is required in order to simulate a subsurface maximum in Hg concentration, as observed in the present-day real ocean. In the steady state, with no anthropogenic enhanced deposition, a somewhat slower Hg sinking flux would still generate a subsurface maximum, but it is harder to have a subsurface maximum at the end of a period of enhanced Hg deposition, as

today.

Figure 9 shows the total ocean inventory of anthropogenic Hg throughout the Anthropogenic deposition period (ending in the year 2100) and beyond, as a function of the Hg particle sinking velocity. Particle transport has only a minor impact on the global rate of Hg uptake during the Anthropocene stage, but strong particle transport has the effect of sequestering the anthropogenic Hg deeper in the ocean (Figure 3), where it is retained somewhat longer than in models with less particle

transport. The model, when forced with an instantaneous end to anthropogenic emissions, predicts that the ocean will continue degassing Hg for 1000 years. When this prediction is turned around, to impose a condition that the Hg deposition rate declines over 1000 years after the year 2100, the duration of the anthropogenic Hg load on the oceans increases to several thousand years.

### 3.2 Model sensitivity to reaction kinetics.

For each of eight kinetic rate constant parameters in the Hg system, we ran simulations to a natural steady state with factor-of-two increases and decreases in each parameter in turn, as shown in Figure 10. In general, increasing the rate constant for a given reaction will increase the concentration of the product, and decrease that of the reactant. The other species' concentrations will also change in the new steady state balance. The concentration of Hg overall depends on the rate of Hg removal from the system, primarily by gas evasion of the minor species Hg(0), with secondary sinks by DMHg evasion and

Hg(II) adsorption onto sinking particles (Table 1). Increasing the rate of MMHg production from Hg(II), for example, decreases [Hg(II)] and increases [MMHg]. The concentration of DMHg increases due to its close coupling with MMHg (see Figure 1).

Changes in the rate constants that produce or consume Hg(0) tend to result in larger changes in Hg concentrations than the rate constants for reactions that involve DMHg, because Hg(0) is responsible for a larger fraction of the gas evasion flux.

The exception is reductive degradation of MMHg to Hg(0), which occurs primarily in the surface ocean, changing the MMHg concentration there without changing concentrations appreciably in the deep ocean. The highest model sensitivity in the suite of runs is to the rate of evasion of Hg(0), which drive large changes in the total Hg concentration of the entire ocean, in the steady state.

In general, the rates of chemical transformation of Hg are much faster than that of ocean overturning circulation, so the

distribution of Hg species at any location reflects a local balance between sources and sinks of each form of Hg. However, reactions at the sea surface that provide a pathway for Hg evasion into the atmosphere have the potential to alter the Hg concentrations throughout the ocean in the steady state.

### 3.2 Isotopic Fractionation

Isotopic fractionations in the Hg cycle can be "expressed" in the isotopic signatures of Hg species, or not, depending on how the fractionating process fits into the network of reactions in the cycle. Figure 11 shows the isotopic compositions of Hg species resulting from a variety of fractionation mechanisms, in schematic diagrams of the ocean Hg cycle. Red numbers indicate isotopic fractionations and black numbers show global mean oceanic isotopic compositions. The results represent pre-anthropogenic steady state. The model is run to equilibrium for each of 7 fractionation mechanisms in isolation, and finally for all mechanisms combined. For ease of comparison with oceanic measurements, all scenarios are subject to fractionation in Hg deposition as indicated by the red numbers next to these fluxes. Figure 12 shows depth profiles of isotopic composition. Figure 13 shows maps of isotopic compositions, at the sea surface and at depth.

A guiding principle in understanding these results is that in the steady state the isotopic composition of the sinking fluxes have to balance the isotopic compositions of the inputs of Hg(II) from rain and Hg(0) from atmosphere-sea surface exchange. A fractionation mechanism that alters the isotopic signature of one of the sink fluxes will require the steady-state signatures of the other sink fluxes to change in compensation. Then the values of the other species, MMHg, and Hg(II), are pulled in various ways by their connections with the two potential gases Hg(0) and DMHg.

### 3.2.1 Gas Evasion Fractionations

The schematic diagram in Figure 11a shows the fractionation associated with evasion of Hg(0) to the atmosphere. The lighter isotope reacts faster (as is typical), leaving a dissolved Hg(0) pool that has residually higher $\delta^{202}$Hg, by about +0.17‰. The Hg(0) evasion flux has a $\delta^{202}$Hg value of -0.23‰, (from the source isotopic composition of 0.17‰ adjusted for the fractionation of -0.4‰), which is lower than the weighted sum of the input fluxes (+0.13‰) and must be balanced by evasion of high $\delta^{202}$Hg DMHg and burial of deep water Hg(II) adsorbed onto particles. The depth dependence of the response is weak (Figure 12a), but variations in particle export from the surface ocean perturb the spatial uniformity of $\delta^{202}$Hg(0) (Figure 13a). Fractionation in DMHg degassing (Figure 11b) is similar in that the $\delta^{202}$Hg of the degassing fractionating species (DMHg) becomes more positive in the residual fraction. Because DMHg is not returned to the Hg pool as quickly in the model as Hg(0), the isotopic deviation in DMHg does not pass to the other pools, which reflect the balance of the source fluxes.

### 3.2.2 Reaction Fractionations

The expression (or not) of a fractionation in a specific reaction pathway in the Hg cycle depends on the web of reactions between the species and the mass balance constraints. For example, fractionation during the reduction step from Hg(II) to Hg(0) (Figure 11c and d) pulls the $\delta^{202}$Hg value of Hg(0) to a lower value, requiring a positive excursion in the $\delta^{202}$Hg value of DMHg to balance the overall evasion isotopic ratio against that of deposition. The $\delta^{202}$Hg values of Hg(II) and MMHg follow DMHg to higher values. This fractionation does lead to surface–deep isotopic contrast in $\delta^{202}$Hg, and regional

variations at the sea surface and in the deep ocean, reflecting differences in particle scavenging in productive areas and the differing fractionation due to photochemistry in the surface ocean. The $\Delta^{199}$Hg isotopic system behaves similarly, with differences due to different isotopic signatures of wet and dry Hg deposition, and with the difference that most of the surface/deep and deep Pacific/Atlantic contrasts in $\Delta^{199}$Hg values can be attributed to this fractionation mechanism alone (Figure 12).

Fractionation in the photochemical MMHg $\rightarrow$ Hg(0) reaction step (Figure 11e and f) causes an increase in the $\delta^{202}$Hg value of MMHg, which is passed on to DMHg and Hg(II). The Hg(0) evasion flux has only slightly lower $\delta^{202}$Hg than the mean deposition flux, balanced by slightly higher $\delta^{202}$Hg in the DMHg evasion and Hg(II) loss on particles. In contrast to Hg(II) reduction, MMHg reduction does not lead to surface/deep $\delta^{202}$Hg contrast in the Hg(II) (Figures 12 and 13), but it does lead to an enrichment in $\Delta^{199}$Hg of MMHg in the surface ocean (Figure 14), consistent with the measurements of fish Hg by Blum et al (2013), and in contrast with the apparent $\Delta^{199}$Hg of Hg(II) in particles from the upper ocean derived from (Motta LC, submitted).

Fractionation in MMHg production (Figure 11g) results in a decrease in $\delta^{202}$Hg of MMHg, forcing $\delta^{202}$Hg values of Hg(II) to increase in compensation. The higher $\delta^{202}$Hg values of Hg(II) on sinking particles offsets slightly lower $\delta^{202}$Hg values of Hg(0) and DMHg evasion to the atmosphere. Fractionation in biologically mediated MMHg degradation (Figure 11h) acts in the opposite sense for $\delta^{202}$Hg values due to the opposite direction of the reaction. Both fractionation mechanisms lead to horizontal heterogeneity of $\delta^{202}$Hg in the surface ocean, and a contrast between deep Atlantic and Pacific values (Figures 12 and 13). These mechanisms do not impact $\Delta^{199}$Hg values because they are purely mass dependent fractionations. Hg(II) adsorption onto particles generates horizontal gradients in $\delta^{202}$Hg of the surface ocean (Figure 13), but in the global mean the fractionation between the surface ocean and deep ocean is very small (Figure 12).

The $\Delta^{200}$Hg isotopic system does not fractionate internally in the ocean, but rather the ocean acts to integrate the isotopic signature of the surface forcing mechanisms, wet and dry deposition. If the deposition is taken to be spatially uniform, the oceanic distribution of $\Delta^{200}$Hg values will also be uniform throughout the ocean (Figure 15). The global mean $\Delta^{200}$Hg values of the ocean might serve as a constraint on the relative magnitudes of the wet and dry fluxes (Figure 16). Regional variations in $\Delta^{200}$Hg values of the ocean may arise from heterogeneity in the deposition fluxes. In Figure 17, the deposition flux of Hg(II) was doubled in each of the Atlantic, Pacific, and Indian basins in turn and the model run to a natural equilibrium. Variations in the $\Delta^{200}$Hg values of deposition into the surface waters of the basin can also be seen, in a muted way, in the $\Delta^{200}$Hg values at 3 km depth in the ocean. However it must be noted that the predicted variations are small (<0.01‰) and with current analytical methods they would be impossible to measure.

## 4 Conclusions

We have embedded a model of Hg chemistry and dynamics into the HAMOCC off-line ocean tracer advection model, including treatment of isotopic fractionation of Hg in the ocean Hg cycle. The efficiency of the model makes it possible to do numerous sensitivity experiments for testing hypotheses and developing intuition about this complex system: 55 simulations of over 10 kyr each are presented in this paper.

The model demonstrates that the Hg cycle in the ocean is closer to an advective end member than to a system in which transport on sinking particles dominates. The interplay of advection by fluid flow and sinking of Hg adsorbed on sinking particles is illustrated by end-member cases in which one or the other dominates. In an advection-dominated case in which particle transport is disabled, the Hg concentration in steady state is relatively uniform with depth, displaying the same pattern as for salinity. In a particle-dominated scenario in which fluid advection of Hg is disabled, the concentration of Hg in steady state increases with depth, in proportion to the decrease in the POC sinking flux with depth (due to particle decomposition). This is because in the steady state in which Hg concentrations are not changing with time, the sinking flux of Hg through the ocean must be the same (on a horizontal average) at all depth levels.

A series of sensitivity runs with different Hg-P sinking velocities shows that the observed present-day subsurface maximum in Hg(II) is a product of Hg sinking on particles and the anthropogenic increase in Hg deposition to the surface ocean. Given the 4-times enhanced Hg deposition flux since about 1850 (Streets et al., 2017), if there were no Hg sinking and subsurface release from particles, the highest Hg concentrations would be at the sea surface today. Anthropogenic Hg sinking on particles does not have a strong impact on the net uptake rate of anthropogenic Hg by the ocean, but if the enhanced rates of Hg deposition were suddenly to return to natural levels, a model with strong Hg sinking takes longer to shed its anthropogenic Hg burden. Since oceanic Hg evasion will be recycled and re-deposited, the ocean system seems poised to buffer the environmental Hg concentration for thousands of years.

We show the sensitivity of the steady-state (pre-Anthropogenic, 1850) Hg species concentrations to eight kinetic rate constants in the aqueous Hg cycle. Allowing a reaction to proceed more quickly than a base case tends to result in more of the product and less of the reactant, but the magnitude of the change, and the impact on the rest of the Hg species and the total Hg concentration, vary widely between the various reactions. Changes to the budget of Hg(0), the evasion of which is the dominant loss mechanism for Hg in surface waters, have a strong impact on the rest of the Hg concentrations. Changes to reactions involved in the MMHg budget have a stronger impact on the Hg cycle than changes to DMHg sources or sinks, because MMHg is kinetically tied more closely to Hg(II).

Isotopic variations in Hg have multiple "dimensions" of fractionation, with mass dependent fractionation produced by most processes, and several forms of mass-independent fractionation produced by photochemical reactions. The Hg cycle in the ocean is complex enough that a model is required to predict the "expression" of isotopic fractionations in processes, on the isotopic signatures of Hg species in the ocean, and on the distribution of variations in those signatures. There is wide variation in the expression of isotope fractionation effects in the isotopic composition of Hg standing stocks. In the model,

surface/deep contrasts in $\delta^{202}$Hg (and $\Delta^{199}$Hg) are due largely to fractionation in the rate of Hg(II) biological+photochemical reduction. This mechanism also generates a contrast between Atlantic and Pacific deep isotopic compositions. The photochemical reduction of MMHg generates a dramatic contrast between the $\Delta^{199}$Hg of MMHg and Hg(II) in the surface ocean, consistent with isotopic measurements of fish (Blum et al., 2013). Multiple mechanisms produced patterns in sea

surface $\delta^{202}$Hg signatures, some creating high $\delta^{202}$Hg excursions in productive areas and some producing low $\delta^{202}$Hg excursions. These mechanisms are, for high $\delta^{202}$Hg excursions in e.g. the equatorial Pacific: MMHg photoreduction, biological MMHg production, and Hg(II) adsorption, and for low $\delta^{202}$Hg excursions: Hg(0) evasion, Hg(II) reduction, biological MMHg production, MMHg biodegradation, and Hg(II) adsorption. The $\Delta^{199}$Hg system in the model is entirely driven by photochemical reactions (Hg(II) and MMHg photo reduction). In reality there may be variations in source input,

but the rates and isotopic signatures of Hg deposition are spatially uniform in the model. Both photochemical mechanisms produce heterogeneity at the sea surface driven by differences in particle export. The only depth contrast in $\Delta^{199}$Hg predicted by the model is for $\Delta^{199}$Hg of MMHg due to MMHg photo-reduction. The $\Delta^{200}$Hg of the ocean on global average in the model reflects the balance of wet vs. dry deposition of Hg (Hg(II) vs. Hg(0)), and regional variations in those rain rates at the sea surface in the model may be weakly represented in the isotopic composition of the deep ocean basins.

**Data Availability**

Fortran source code and required execution files are given in the supplemental material and archived along with model output data files at https://doi.org/10.6082/ngqr-zf89.

**Acknowledgements**

This work stands on the shoulders of Ernst Maier-Reimer who created the HAMOCC model. It also benefitted immensely

from the constructive criticism of Jeroen Sonke and another anonymous reviewer.

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

Table 1.  Fluxes in Mmol/yr from model kinetic rate constant sensitivity experiments.

| Experiment | Hg(0) Evasion | DMHg Evasion | HgP surf | HgP seafloor |
|---|---|---|---|---|
| Base | 3.69 | 0.07 | 1.12 | 0.57 |
| Hg(II) -> MM Hg 2x | 3.77 | 0.08 | 1.03 | 0.47 |
| 0.5x | 3.65 | 0.06 | 1.17 | 0.64 |
| MMHg -> Hg(II) 2x | 3.67 | 0.06 | 1.14 | 0.63 |
| 0.5x | 3.71 | 0.07 | 1.10 | 0.49 |
| Hg(II) -> Hg(0) 2x | 3.75 | 0.06 | 1.06 | 0.43 |
| 0.5x | 3.66 | 0.07 | 1.15 | 0.66 |
| Hg(0) -> Hg(II) 2x | 3.68 | 0.07 | 1.13 | 0.66 |
| 0.5x | 3.70 | 0.07 | 1.11 | 0.46 |
| MMHg -> Hg(0) 2x | 3.72 | 0.06 | 1.10 | 0.56 |
| 0.5x | 3.66 | 0.08 | 1.14 | 0.59 |
| Hg(II) -> DMHg 2x | 3.66 | 0.12 | 1.10 | 0.53 |
| 0.5x | 3.71 | 0.04 | 1.12 | 0.62 |
| MMHg -> DMHg 2x | 3.70 | 0.07 | 1.12 | 0.62 |
| 0.5x | 3.69 | 0.07 | 1.12 | 0.50 |
| DMHg -> MMHg 2x | 3.68 | 0.08 | 1.11 | 0.53 |
| 0.5x | 3.70 | 0.06 | 1.12 | 0.60 |
| Hg(0) Evasion 2x | 4.14 | 0.04 | 0.70 | 0.40 |
| 0.5x | 2.96 | 0.10 | 1.81 | 1.06 |
| DMHg Evasion 2x | 3.65 | 0.06 | 1.17 | 0.67 |
| 0.5x | 3.65 | 0.06 | 1.17 | 0.67 |

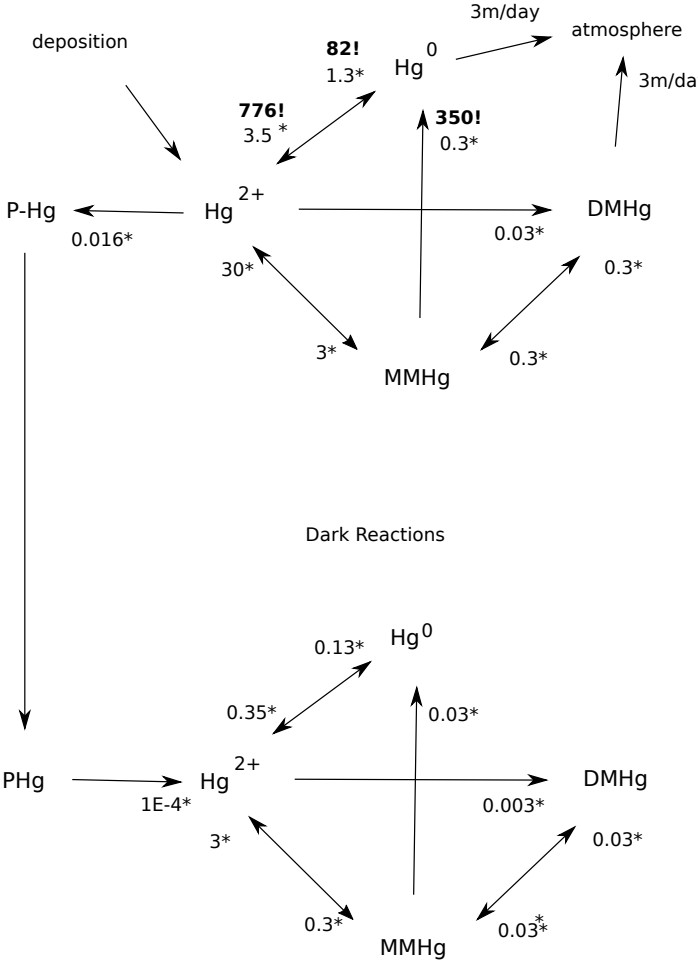

**Figure 1. Schematic of the reaction web for Hg speciation in the model. Starred numbers by the arrows show typical values for biologically mediated rate constants, in units of yr⁻¹. Photochemical rate constants are denoted by bold text and exclamation marks (!). Gas evasion rate constants are based on an ocean-average piston velocity of 3 m/day (Broecker and Peng, 1974). Biological rate constants are fit to (Semeniuk and Dastoor, 2017) and (Zhang et al., 2014a), based on first-order degradation kinetics for POC, resulting in a scaling $k_{bio} = 10^{-6}$ [POC](mol/L). Photochemical MMHg degradation rate constant is from (Bergquist and Blum, 2007). Hg(II) photoreduction and Hg(0) photooxidation rate constants are fits to the global budget from (Soerensen et al., 2010).**

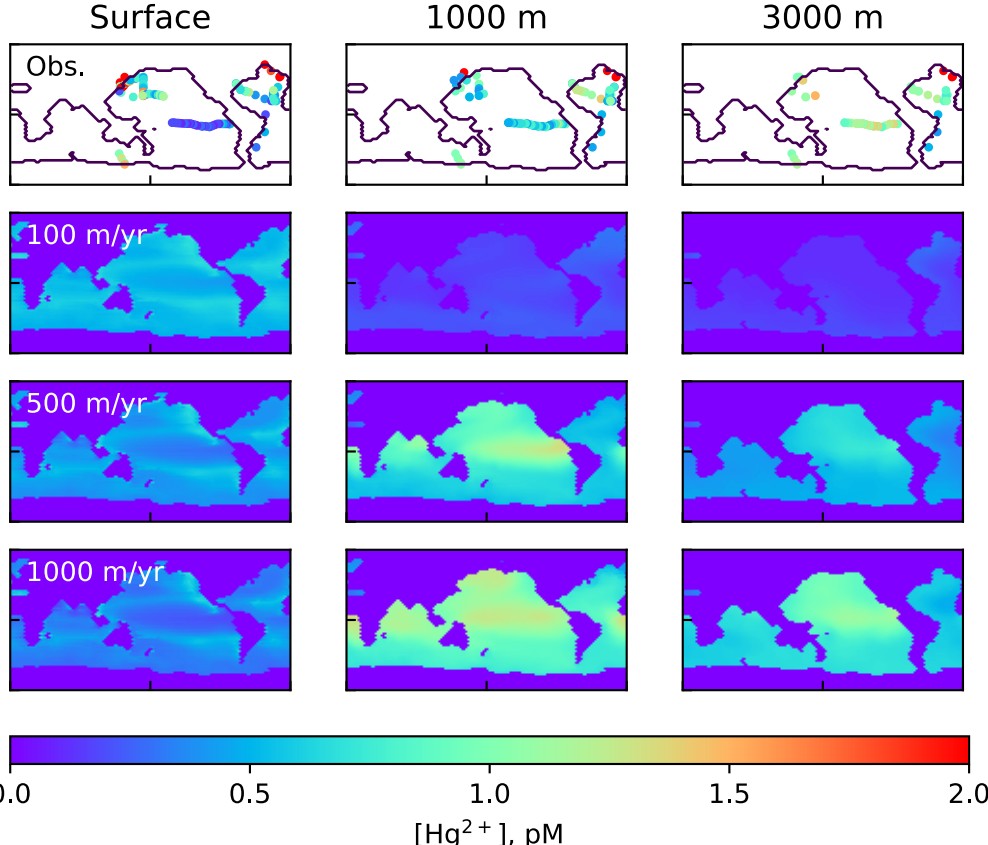

**Figure 2.** Comparison of Hg(tot) concentrations in pM from the model with data in the top row, from (Laurier et al., 2004;Bowman et al., 2015;Bowman et al., 2016;Cossa et al., 2004;Cossa et al., 2011;Hammerschmidt and Bowman, 2012;Lamborg et al., 2012;Mason et al., 2001;Mason et al., 1998), at approximately the depths in the ocean given at the top. Lower three rows are present-day (year 2010) model results using different values of the particulate-bound Hg sinking velocity as indicated by the labels on the left, with 500 m/yr as the the base case.

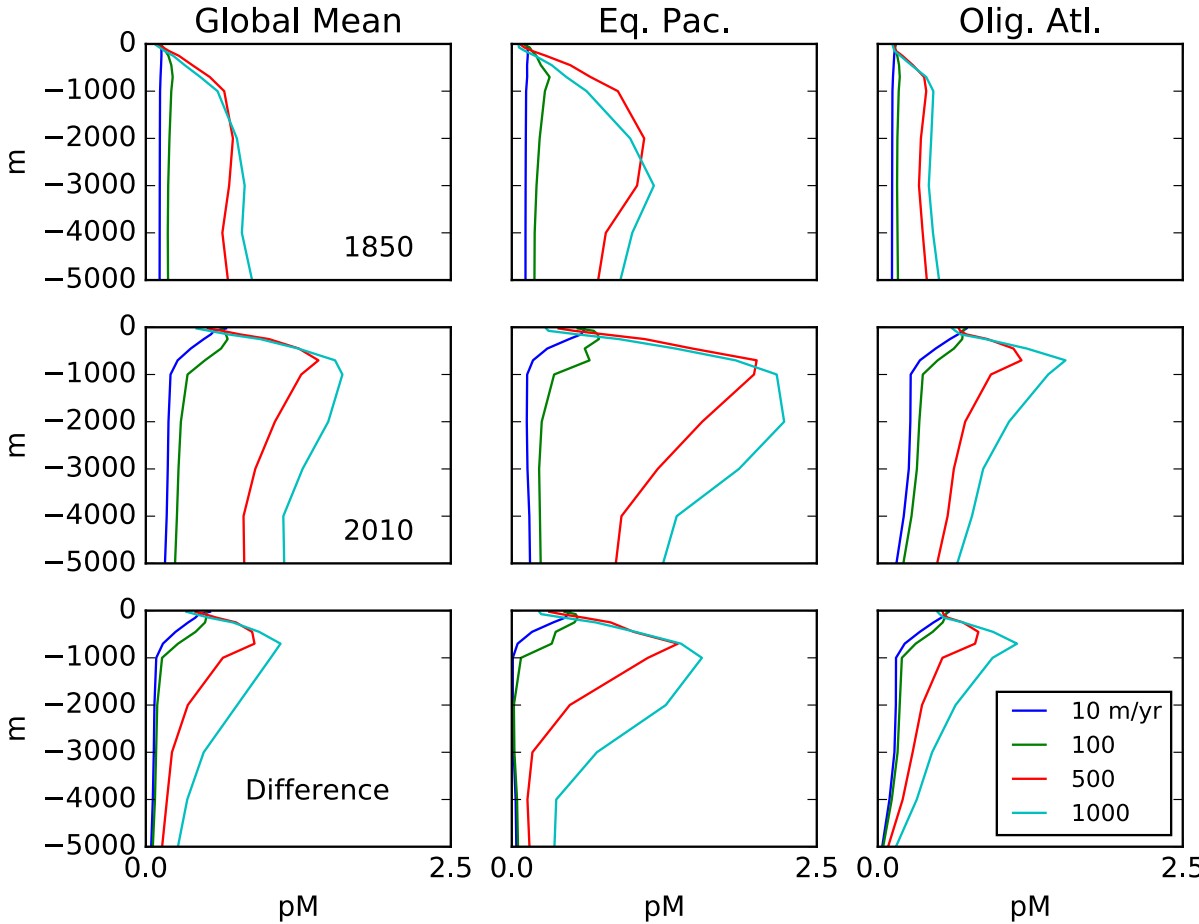

**Figure 3. Depth profiles (in meters) of the total Hg concentration in the model, global mean and from locations shown on Figure 4, preanthropogenic (1850), present-day (2010), and the difference between the two, for different values of the particulate-bound Hg**
10 **flux in m/yr (the base case is 500 m/yr).**

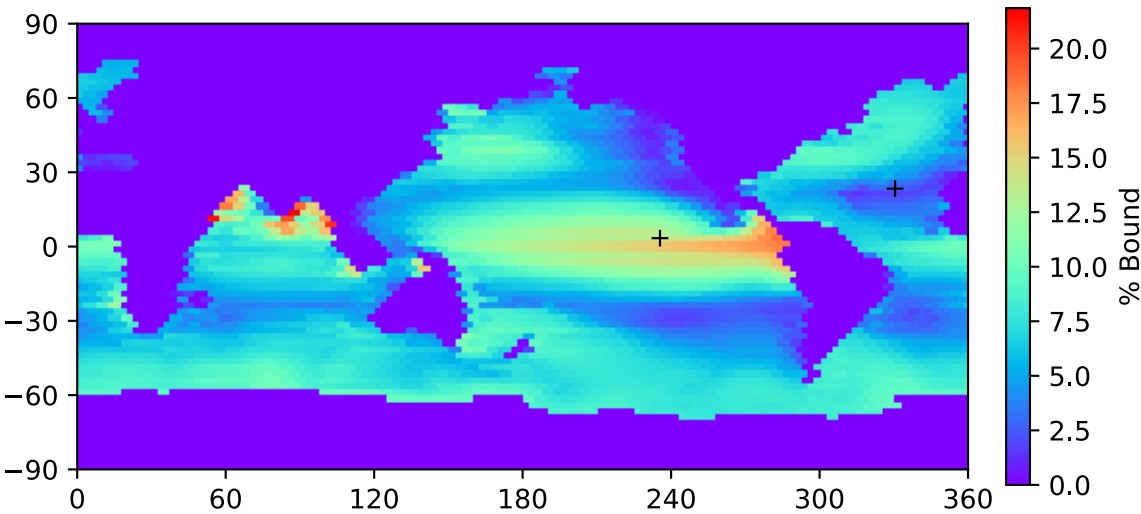

**Figure 4. A map of the bound fraction of Hg(II) (relative to bound + unbound), at the sea surface, when the $K_d$ value is $2 \cdot 10^6$. Higher POC concentrations in productive regions lead to higher bound fractions of the Hg(II). Plusses indicate the locations of profiles in Figures 3, 8, 12, 14, and 15.**

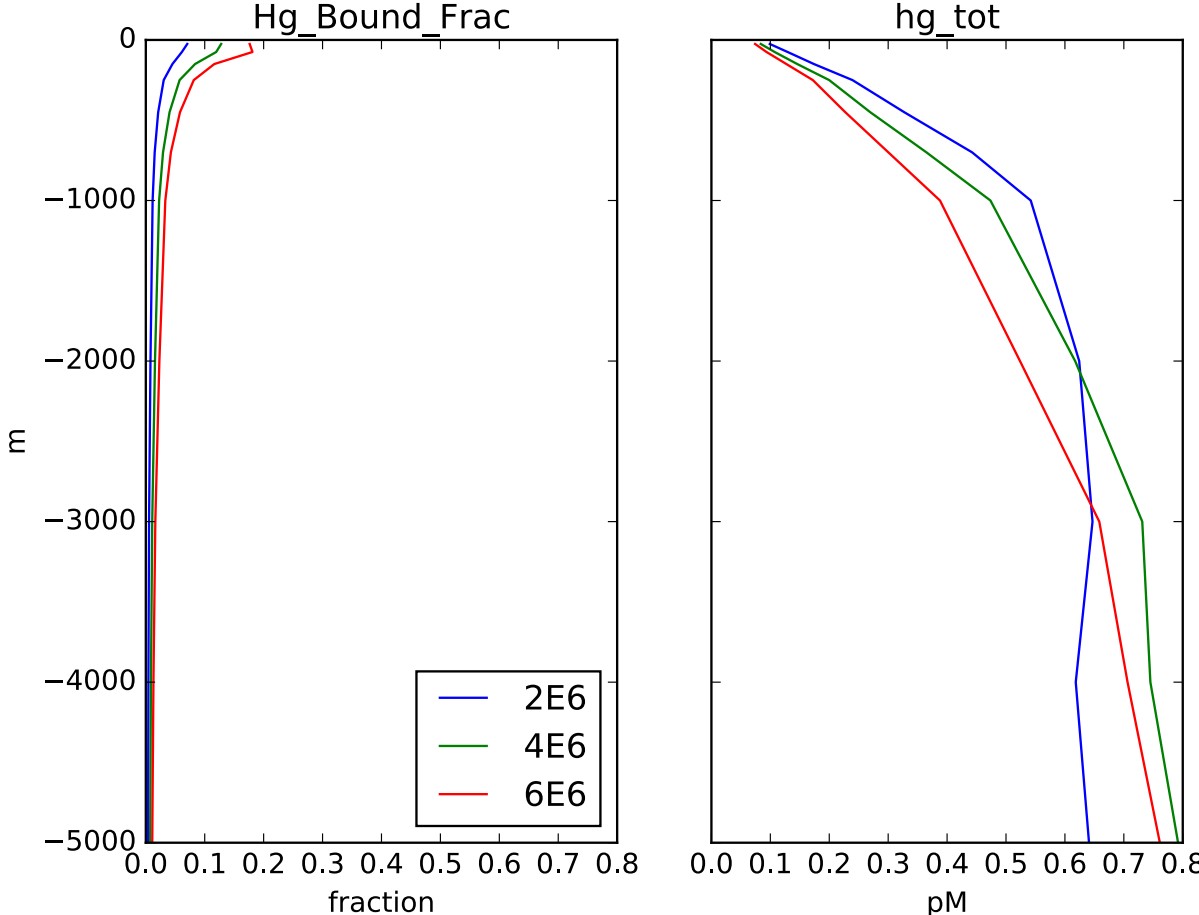

**Figure 5.** Depth profiles (in meters) of the bound fraction of Hg(II) from the same locations as in Figure 3, as a function of the $K_d$ value, preanthropogenic steady state (1850). A value of $2 \cdot 10^6$ l/kg POC (blue lines) is used in the rest of the model simulations.

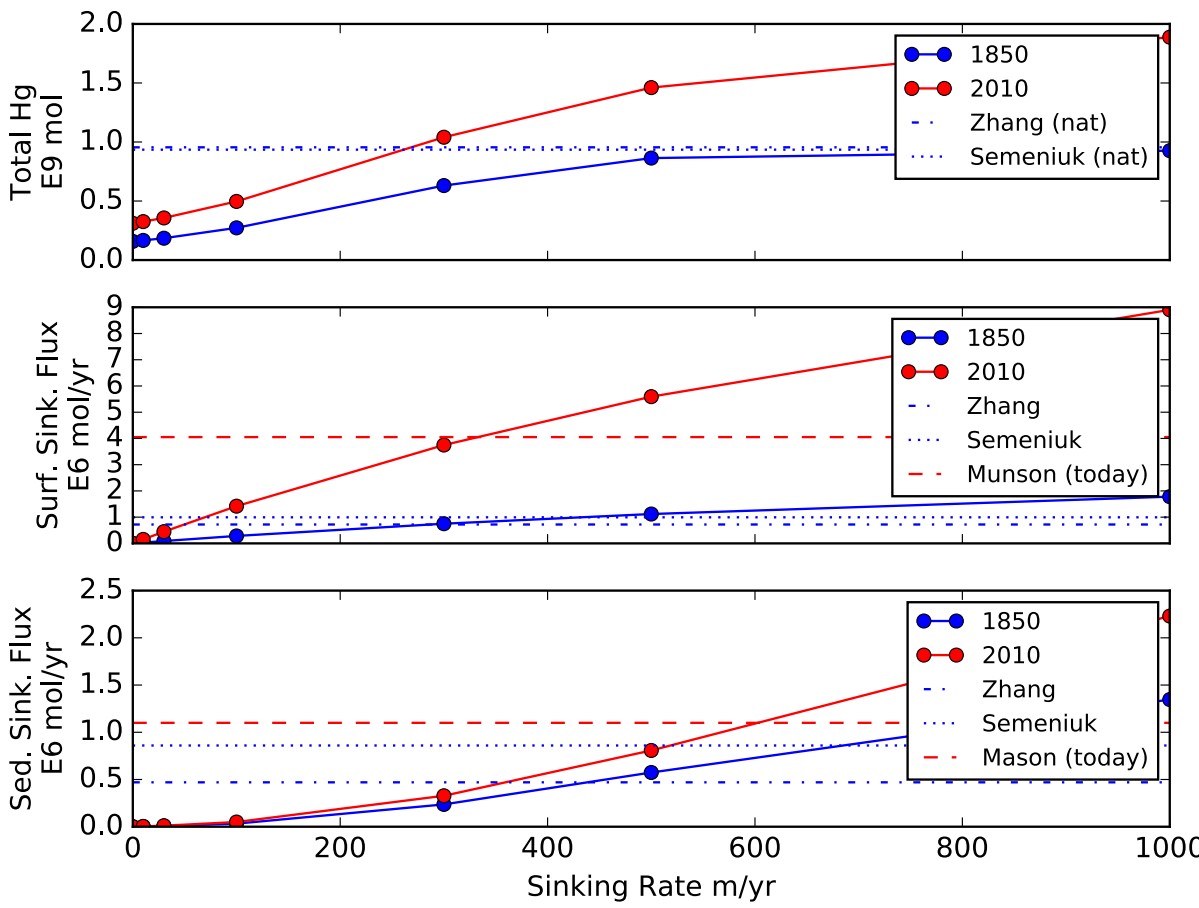

**Figure 6. Global model fluxes as a function of Hg(II) sinking velocity imposed in the model. Colors represent preanthropogenic and present-day results from our model. For comparison broken lines are model results from (Zhang et al., 2014a) and (Semeniuk and Dastoor, 2017), sediment trap data from (Mason et al., 2012), 17° N latitude in the Pacific and 60 m water depth, extrapolated globally. A default value of 500 m / year is the base case for the rest of the simulations in this paper.**

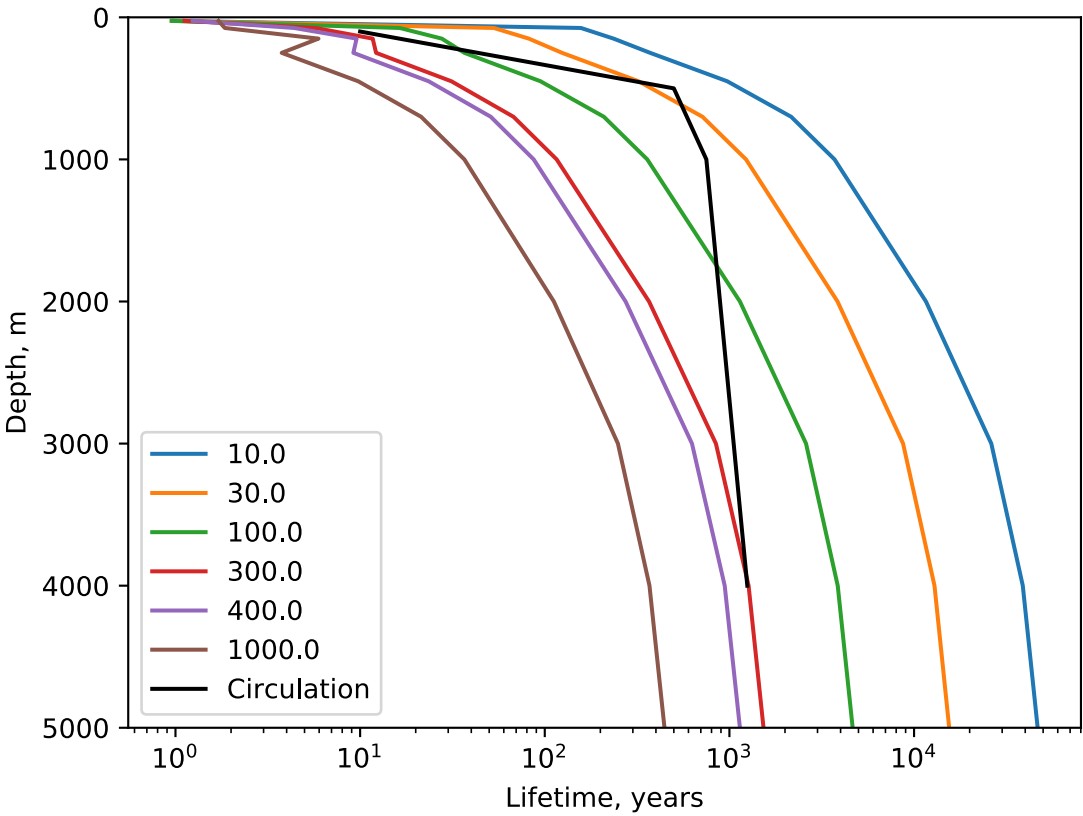

**Figure 7. Profiles of the turnover time of Hg(II) with respect to transiting through the system on sinking particles, as a function of the sinking velocity in the legend, in m/yr. The black line is the water age since exposure to the atmosphere derived from the $^{14}$C distribution (Gebbie and Huybers, 2012).**

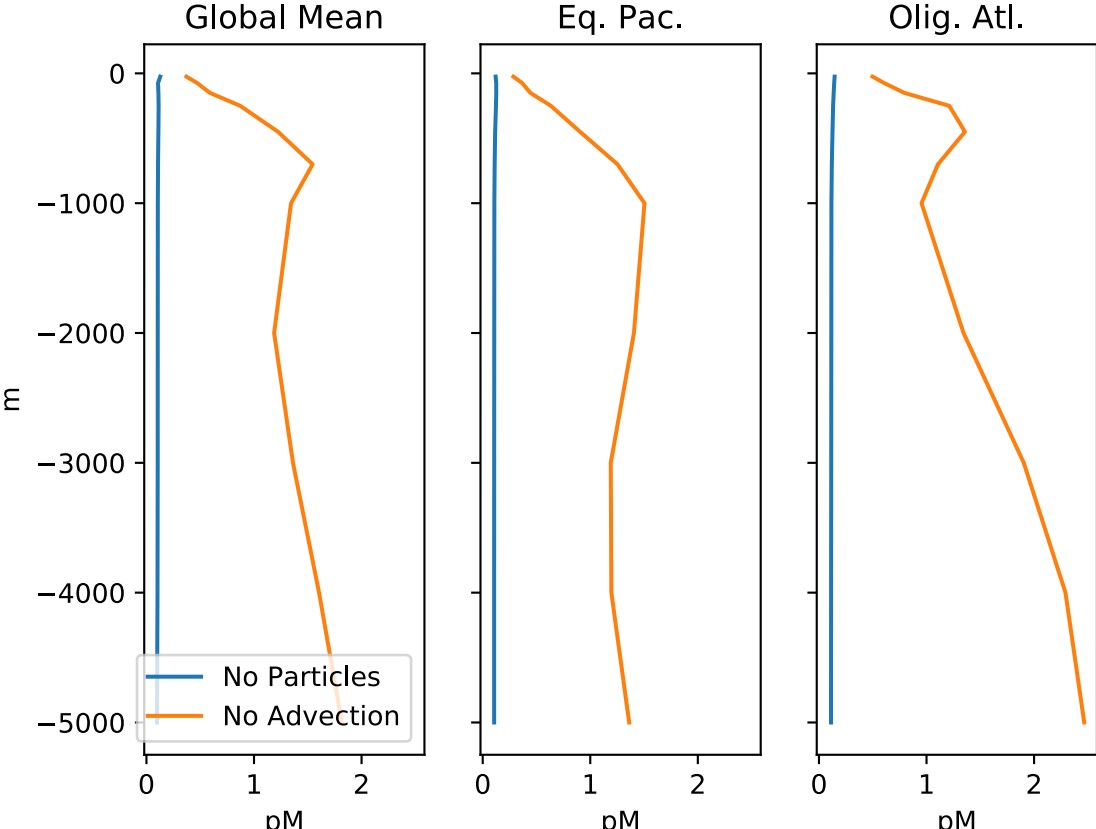

**Figure 8.** Profiles of mean Hg concentration in (preanthropogenic) steady state, as a function of depth in the ocean, in equilibrium, for the end-member cases of no particles (blue lines), and no advection (orange lines). Left panels are global mean, others are from locations in Figure 4. If there were no particles, the Hg concentration would be homogenized throughout the ocean by the circulation. If there were no circulation, the concentration in the steady state increases with depth in the ocean, because there are fewer sinking particles at depth, so the Hg abundance per particle has to increase, as does therefore the dissolved Hg concentration in the ocean.

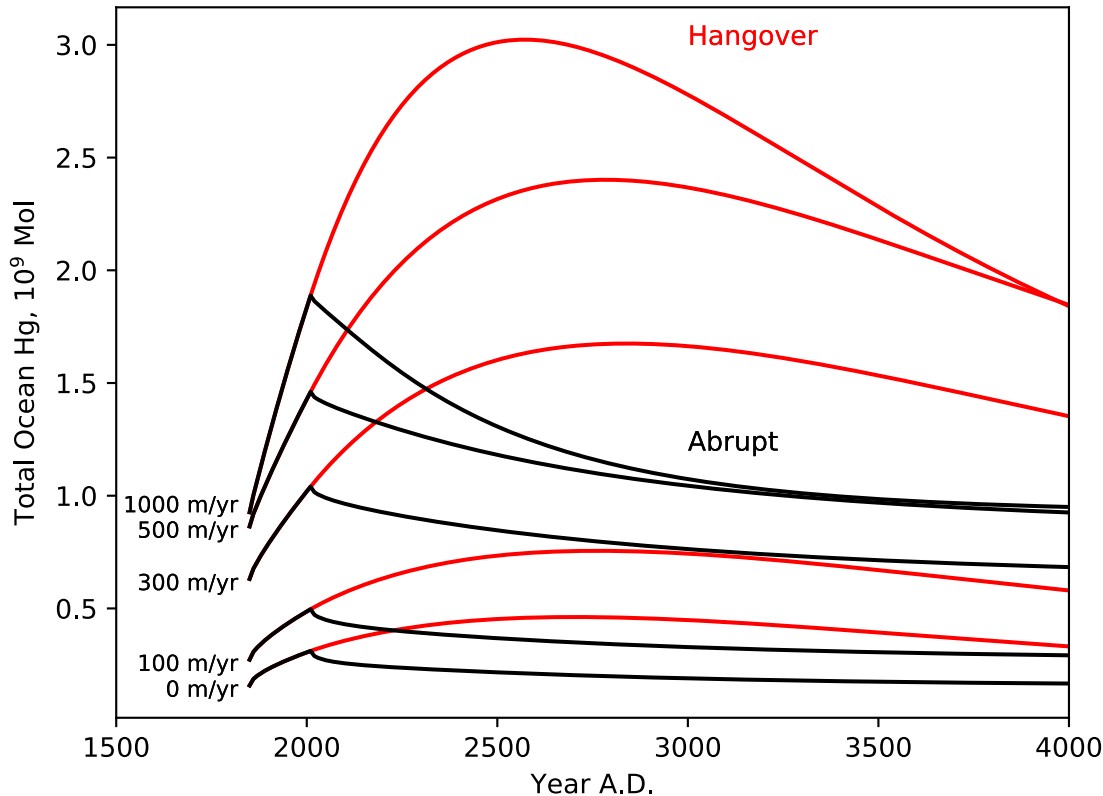

**Figure 9. Time series of the ocean load of Hg (Mmol), in response to 250 years of enhanced Hg(II) deposition (1850-2100), followed by abrupt return to natural Hg deposition rates, or 1000-year wind-down in anthropogenic deposition due to recycling from the ocean.**

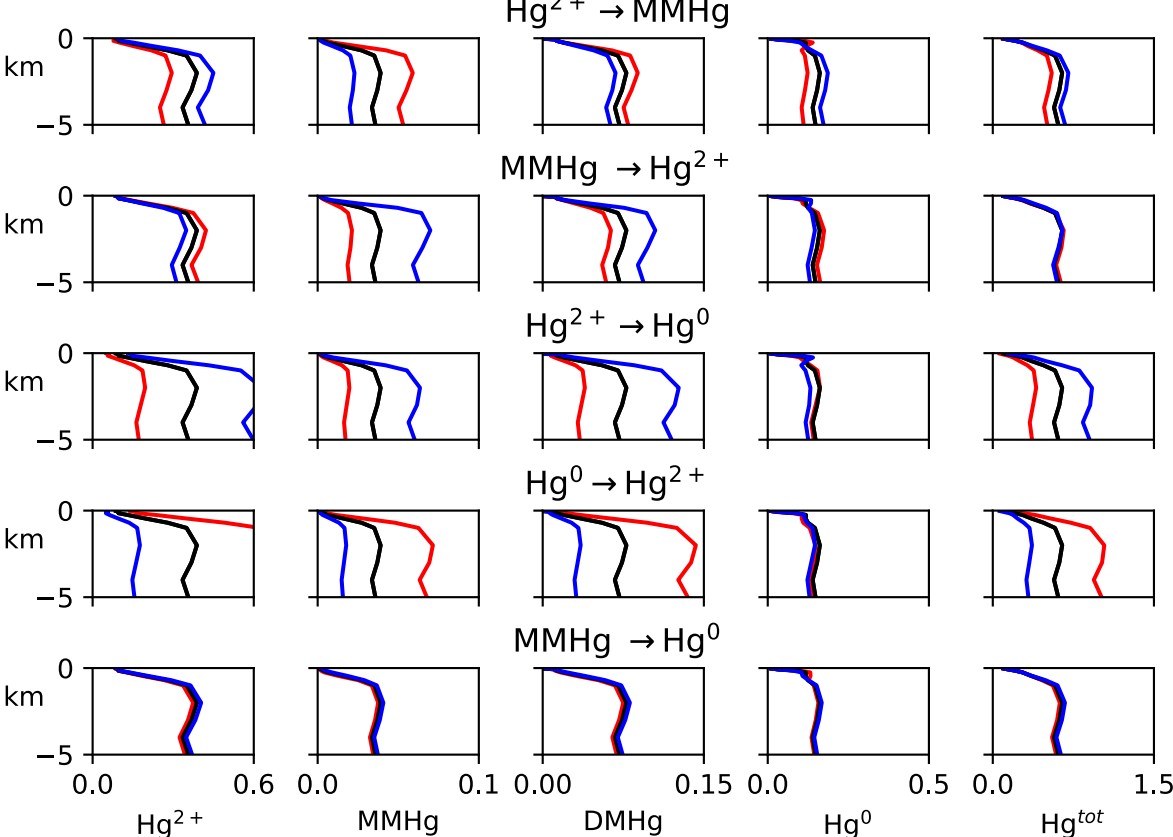

**Figure 10. Model sensitivity to kinetic rate constants in the Hg system. For each kinetic rate constant printed on the left-hand side, global mean concentrations of each species are given in the four plots in that row, as indicated by the labels at the top of each column. Black lines represent the base case, and red and blue represent factors of 2 higher and lower for that kinetic rate constant, respectively.**

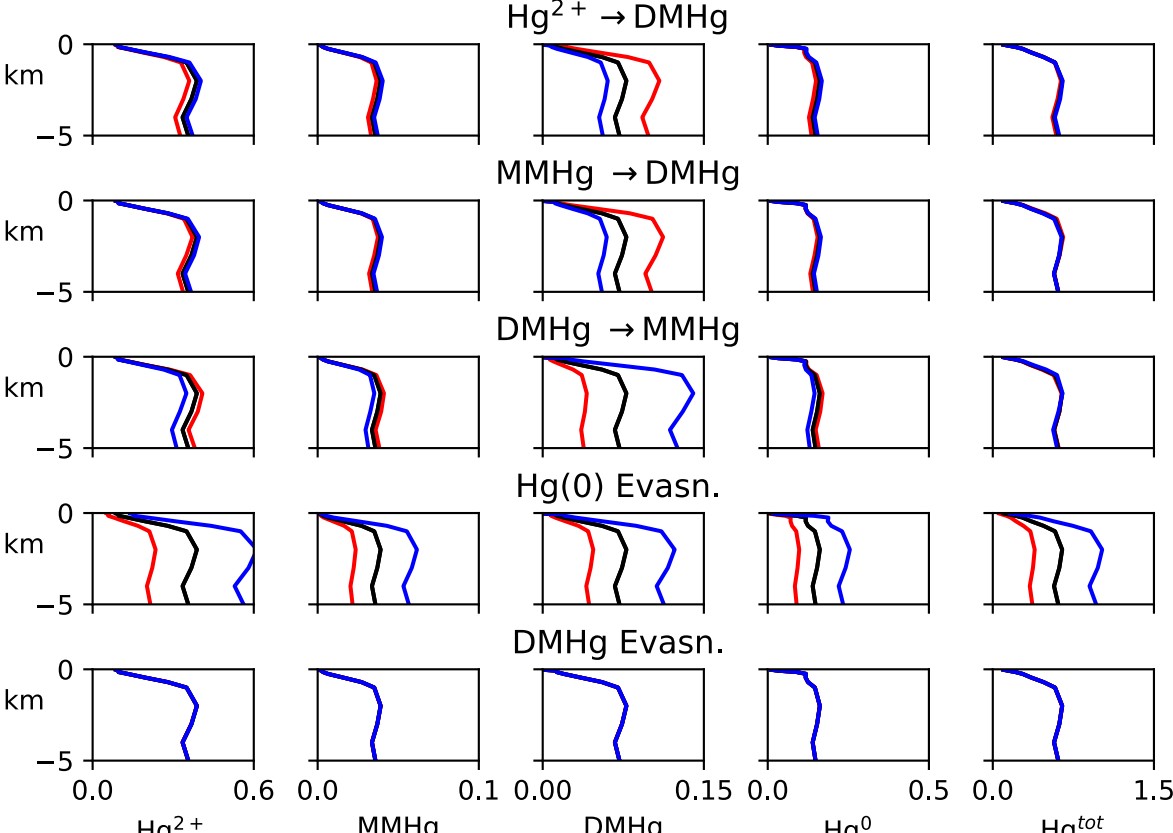

**Figure 10.** Continued

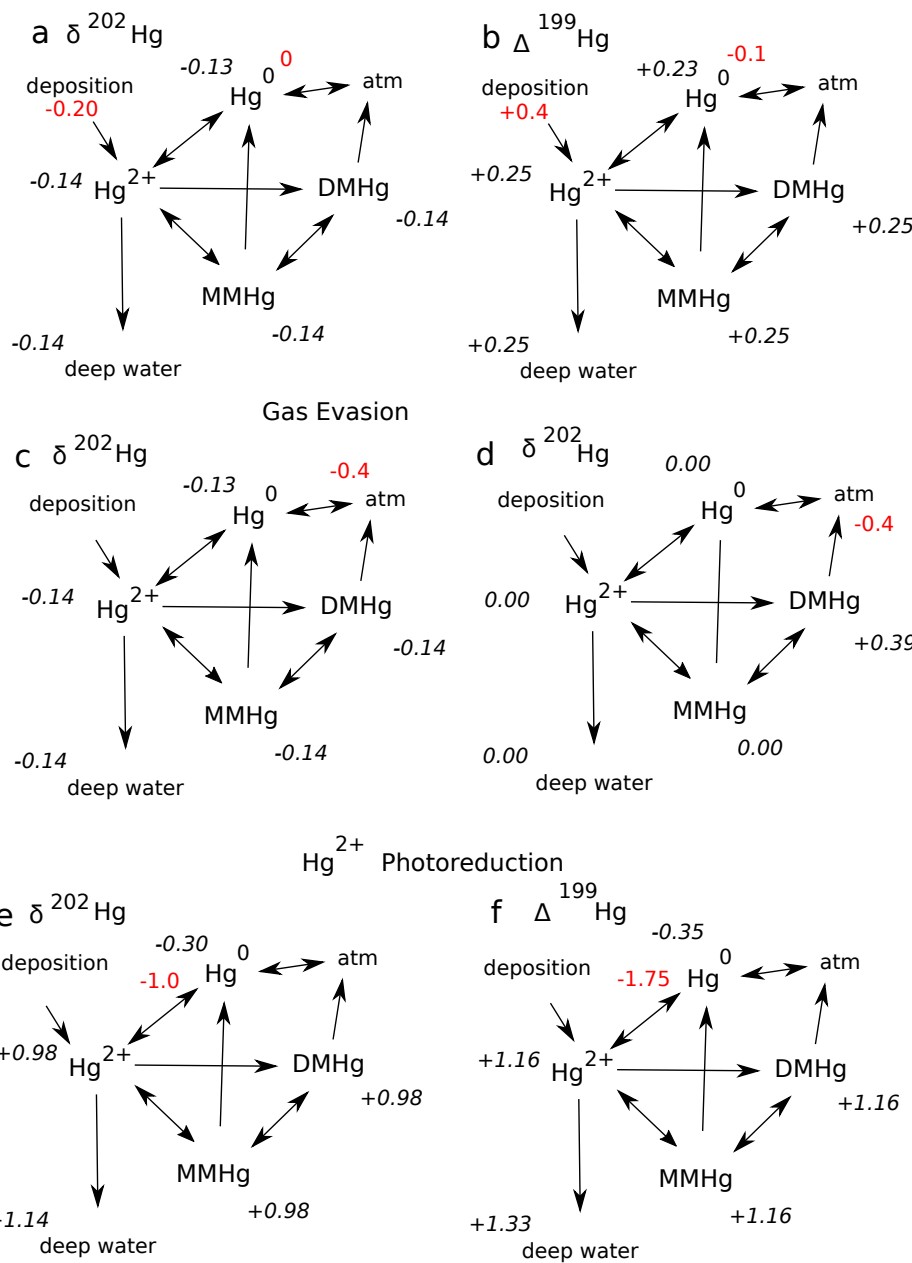

**Figure 11. Schematic of the expression of isotopic fractionations on the global mean sea surface isotopic signatures of the Hg species in the model, for preanthropogenic steady state. Fractionation epsilon values are shown in red, expressed as permil**

differences in the $^{202}Hg/^{198}Hg$ ratio. Resulting global surface average $\delta^{202}Hg$ values for each species are in black italics. Isotopic compositions of the wet and dry deposition are preanthropogenic estimates from Sun et al. {, 2016 #9} A) The signature of mass dependent fractionation in $Hg^{2=}$ deposition.. B) Mass independent $\Delta^{199}Hg$ signature of wet deposition. C) Fractionation associated with Hg(0) evasion (Wiederhold et al., 2010). D) Fractionation applied to DMHg evasion (assuming the same fractionation as for Hg(0) evasion). E and F) Fractionation is applied in the reduction of Hg(II) to form Hg(0) (Kritee et al., 2007). G and H) Fractionation in biological and photo demethylation/reduction of MMHg to form Hg(0). The $\varepsilon^{202}$ isotopic fractionation is taken to be a weighted average of biological (Kritee et al., 2007) and photochemical (Bergquist and Blum, 2007;Blum et al., 2014), while the $\varepsilon^{199}$ is from (Bergquist and Blum, 2007). G) Fractionation of biological MMHg production from (Rodriguez-Gonzalez et al., 2009). H) Biological demethylation from (Kritee et al., 2007). I) Fractionation during adsorption of Hg(II) onto POC is from (Wiederhold et al., 2010).

## Photo Demethylation

g  δ$^{202}$Hg

deposition  *-0.01*  Hg$^0$  → atm
*0.01*  Hg$^{2+}$  −1.0  → DMHg
*0.12*
MMHg
*0.03*  *0.91*
deep water

h  Δ$^{199}$Hg

deposition  *-0.02*  Hg$^0$  → atm
*0.03*  Hg$^{2+}$  −2.5  → DMHg
*0.29*
MMHg
*0.08*  *+2.18*
deep water

## Bio Demeth.

i  δ$^{202}$Hg  *0.00*

deposition  Hg$^0$  ← → atm
*-0.01*  Hg$^{2+}$  −0.4  → DMHg
*0.00*
MMHg
*-0.03*  *0.05*
deep water

## Adsorption onto particles

j  δ$^{202}$Hg  *0.13*

deposition  Hg$^0$  ← → atm
*0.14*  Hg$^{2+}$  → DMHg
−0.6  *0.14*
MMHg
*0.14*  *0.14*
deep water

## All Fractionations

k  δ$^{202}$Hg  *-0.07*  −0.4
deposition  0.4  −1.7  Hg$^0$ 0  0  ← → atm  −0.4
−0.4/1
*1.51*  Hg$^{2+}$  → DMHg
−0.6  *+1.71*
−0.4
MMHg
*1.90*  *-0.04*
deep water

δ

l  Δ$^{199}$Hg  *-0.14*  0
deposition  −1.0  Hg$^0$ 0  ← → atm
0.05  −2.5
*1.44*  Hg$^{2+}$  → DMHg
*+1.69*
MMHg
*1.65*  *+3.59*
deep water

**Figure 11. Continued.**

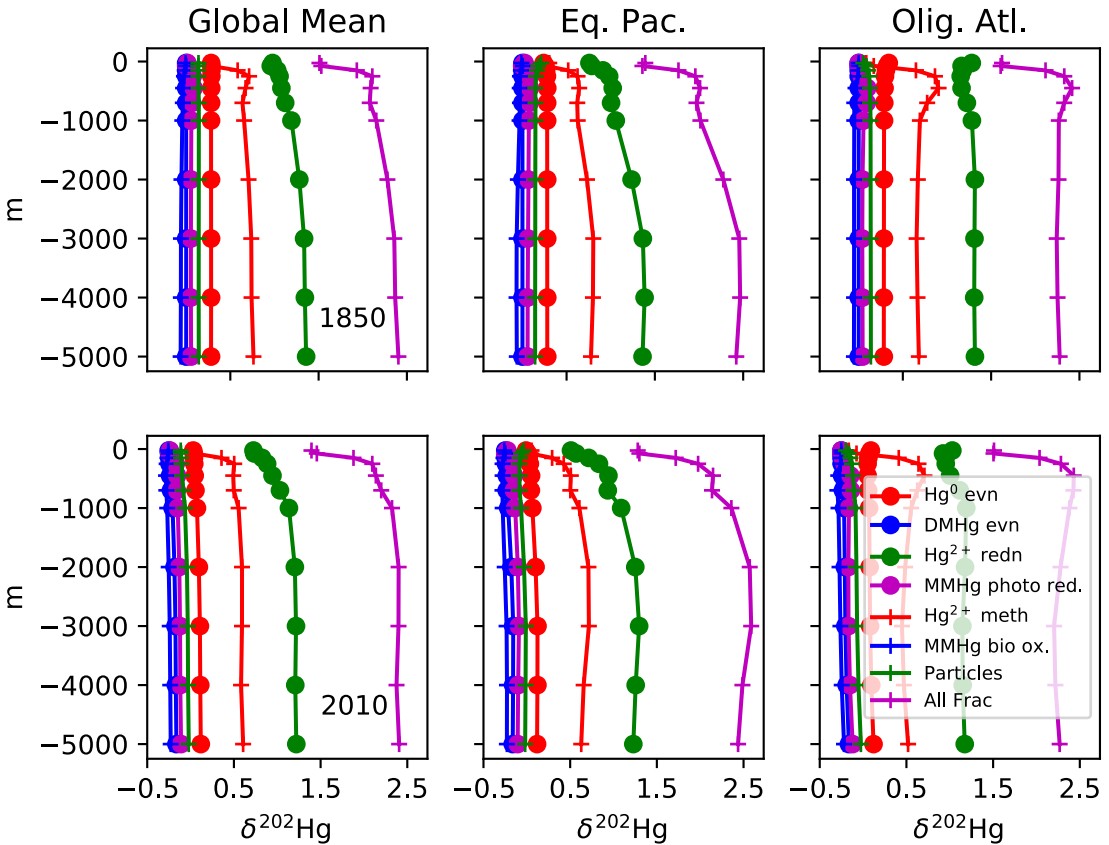

12. Profiles of $\delta^{202}$Hg(II) and $\Delta^{199}$Hg(II) for different fractionation scenarios, global mean, and for the locations in Figure 4.

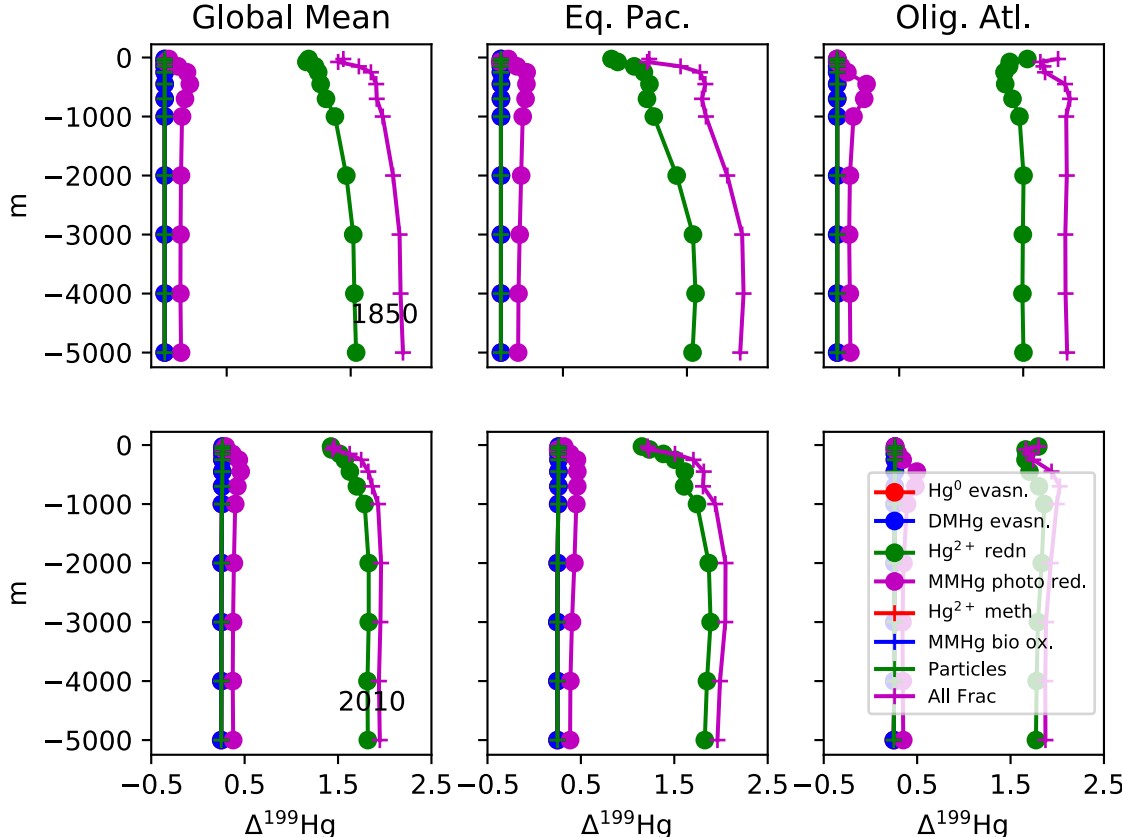

Figure 12. Continued.

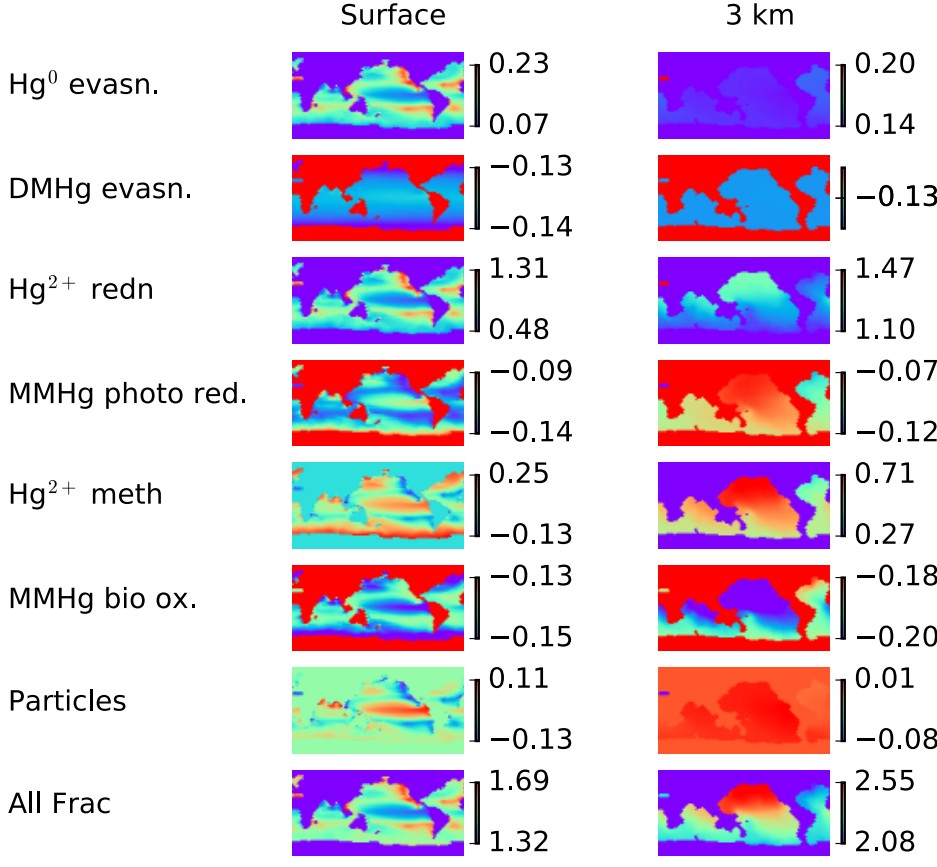

**Figure 13a.** Maps of $\delta^{202}$Hg(II) at the sea surface (left) and at 3 km depth (right) for different fractionation scenarios.

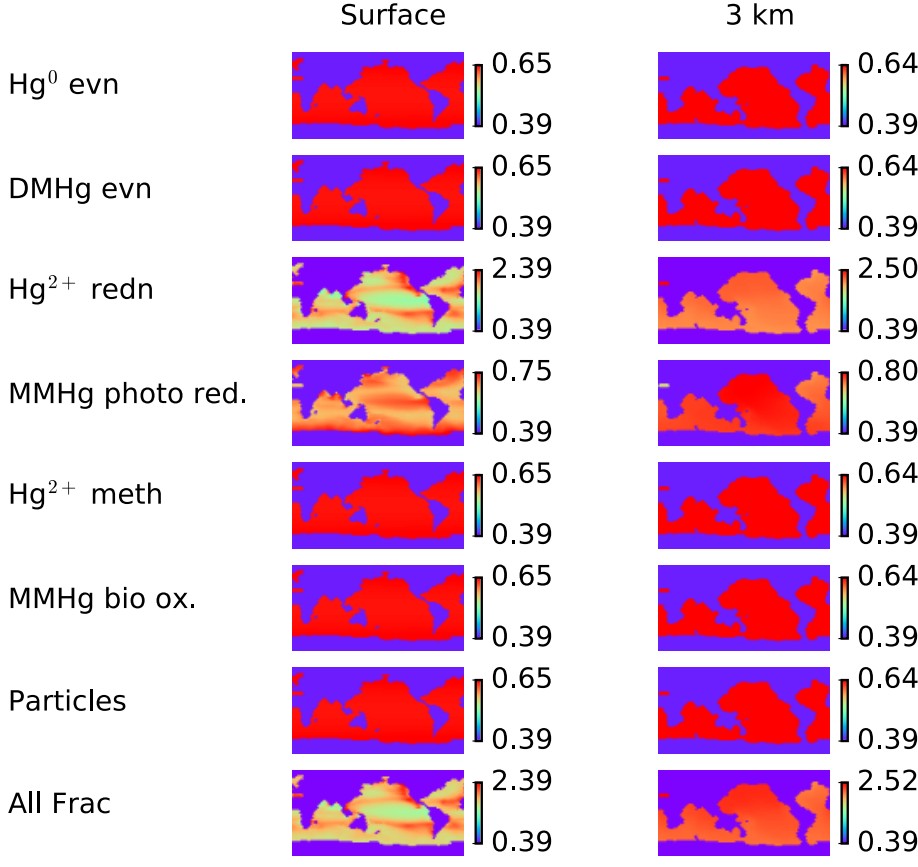

**Figure 13b. Maps of $\Delta^{199}$Hg(II) at the sea surface (left) and at 3 km depth (right) for different fractionation scenarios.**

.

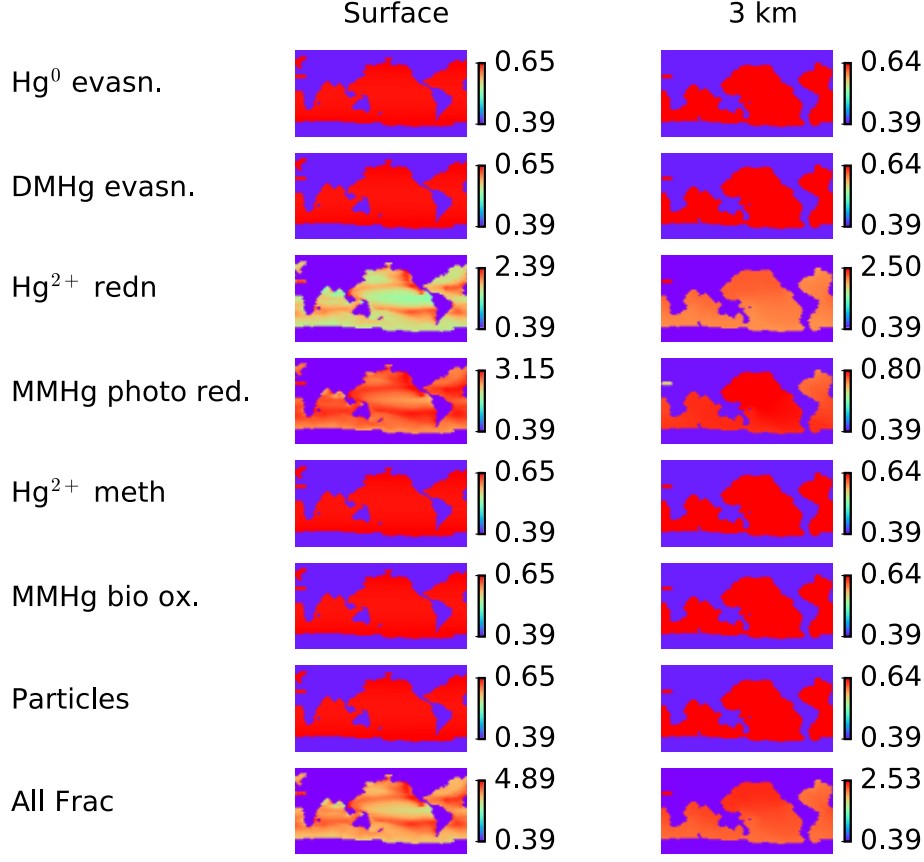

**Figure 13c. Maps of $\Delta^{199}$Hg of MMHg at the sea surface (left) and at 3 km depth (right) for different fractionation scenarios.**

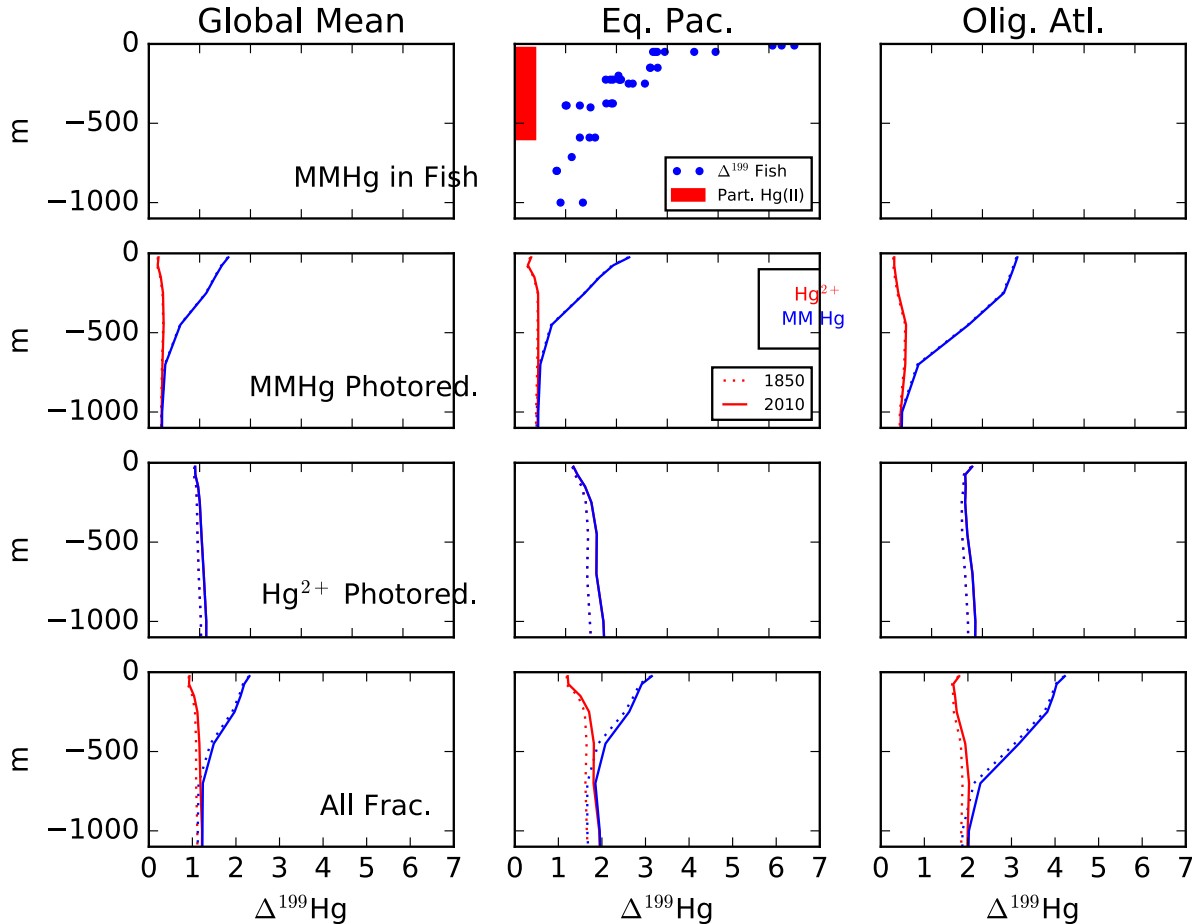

**Figure 14.** Profiles of $\Delta^{199}$Hg of MMHg and Hg(II) for photochemical fractionation mechanisms. Top row is observations of MMHg $\Delta^{199}$Hg inferred from measurements of fish (Blum et al., 2013, and $\Delta199$Hg for Hg(II) from measurements of Hg on particles [Motta LC, submitted #6704). Second row shows model results with MMHg photoreduction, using isotope fractionation from (Bergquist and Blum, 2007). Third row shows the impact of Hg(II) photo reduction (Kritee et al., 2007). Bottom row shows sum of MMHg and Hg(II) photo reduction mechanisms. Locations for the Eq. Pac. and Olig. Atl. profiles are given in Figure 4.

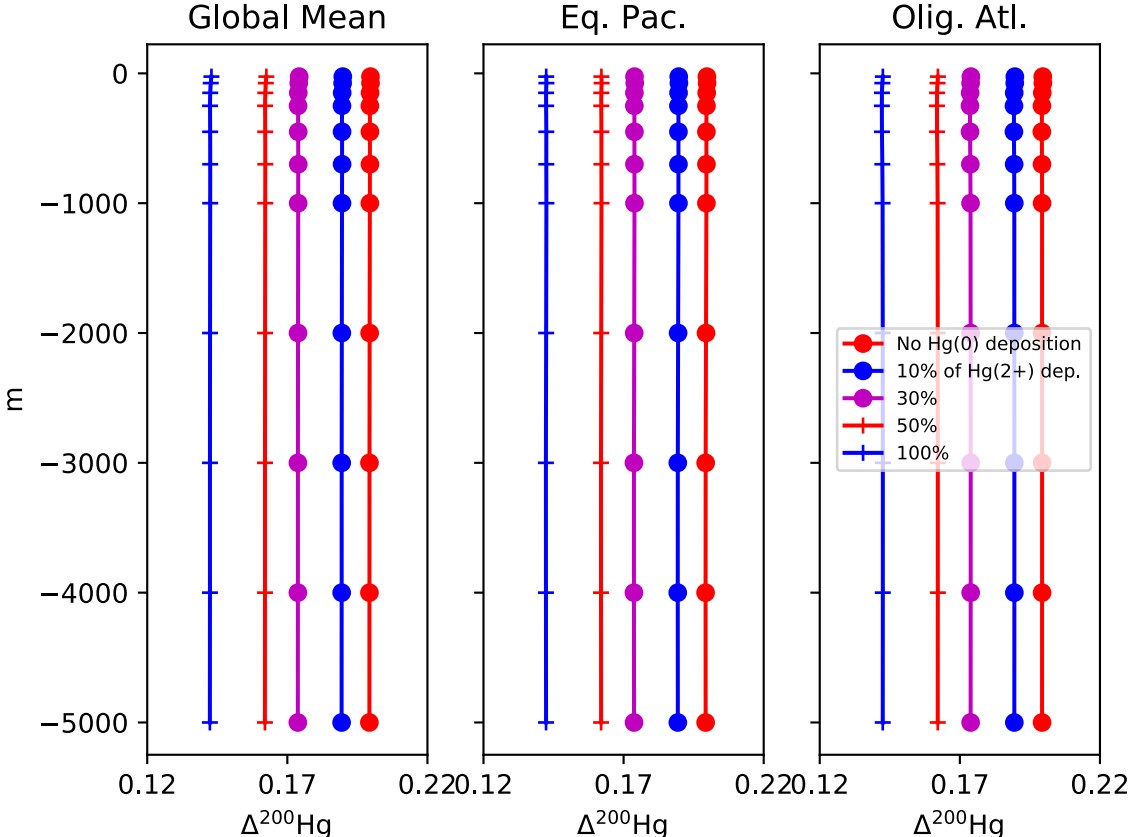

**Figure 15. Profiles of the Δ$^{200}$Hg isotopic composition of Hg(II) for different values of the dry deposition (Hg(0)) flux. The rate of wet deposition is the same for all runs. Locations for the Eq. Pac. and Olig. Atl. profiles are given in Figure 4.**

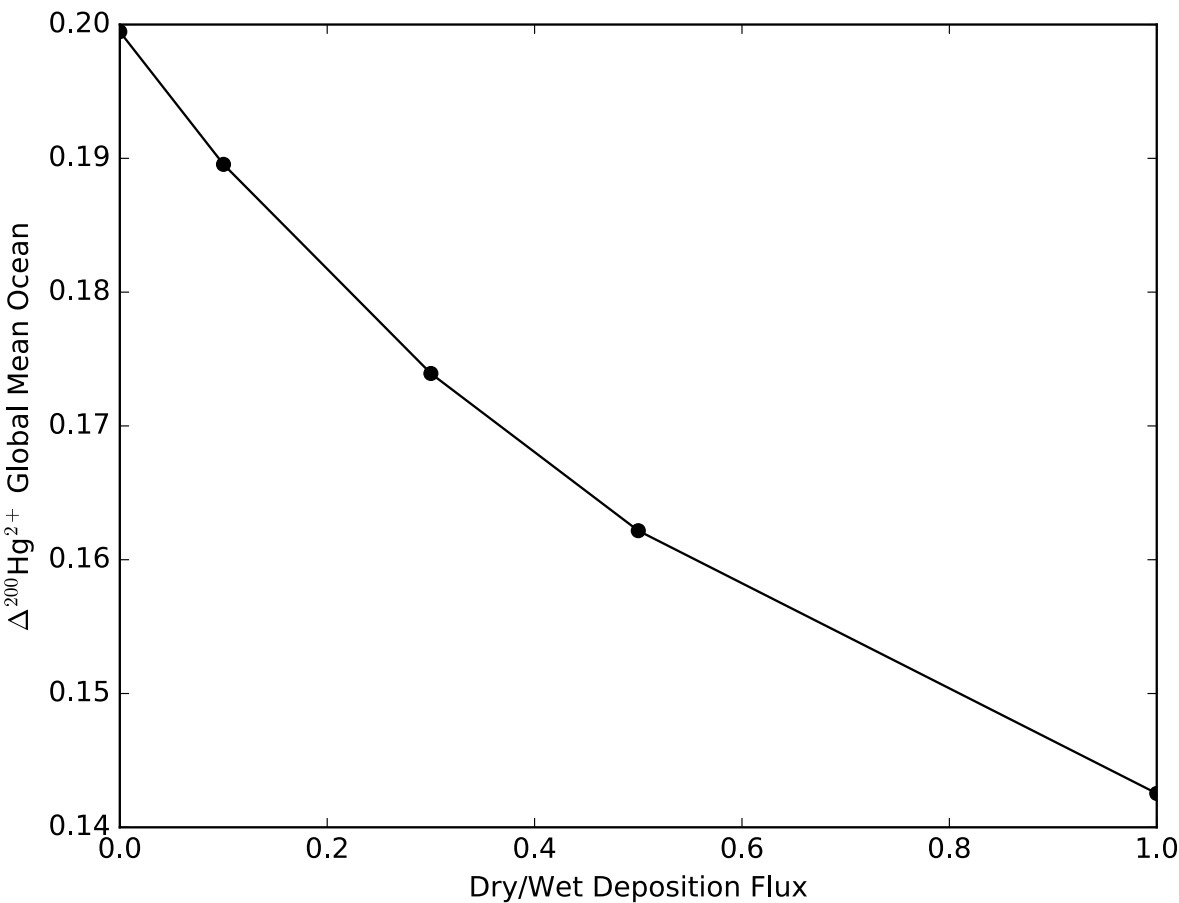

**Figure 16. The $\Delta^{200}$Hg isotopic composition of the global mean ocean is a function of the ratio of wet and dry deposition fluxes.**

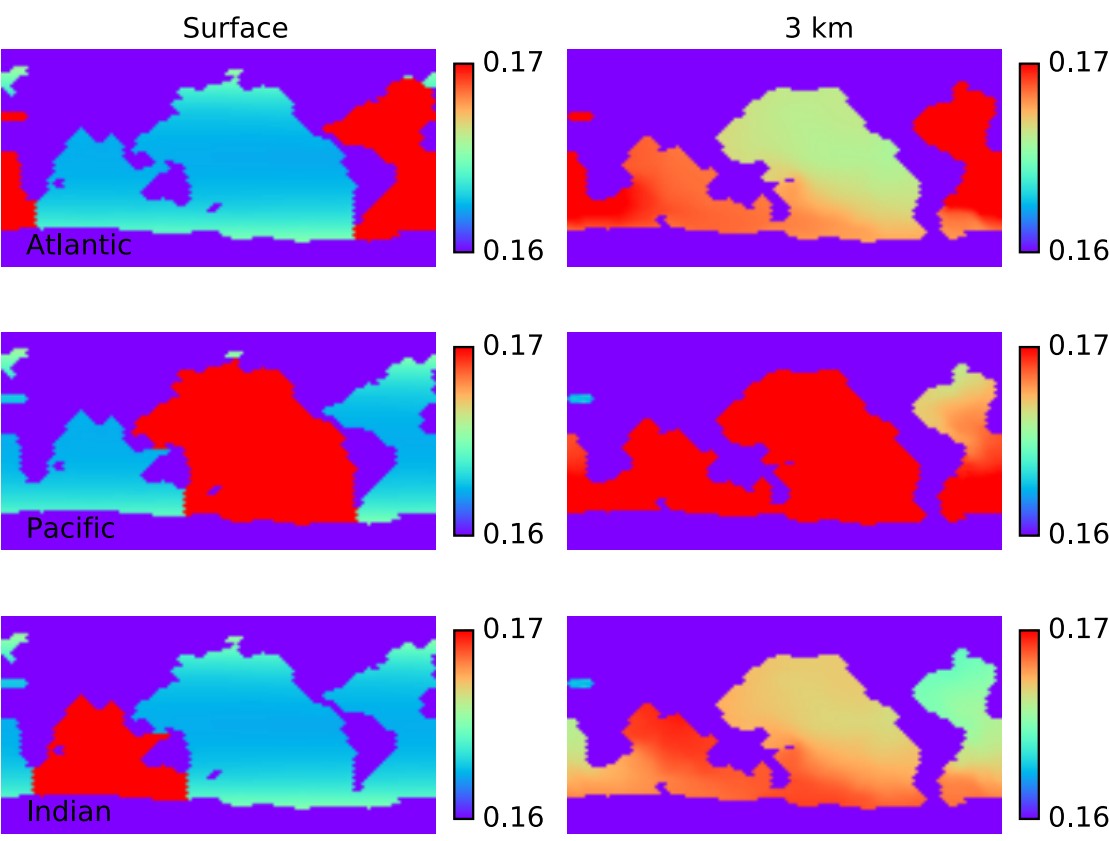

**Figure 17.** Maps of steady-state distribution of $\Delta^{200}$Hg(II) under conditions of a doubling of the Hg(II) deposition flux which is isolated to the Atlantic, Pacific, and Indian oceans. The isotopic signature at 3 km depth reflects differences in surface deposition.

