# Peer review of "A model of mercury cycling and isotopic fractionation in the ocean"

_Biogeosciences, 2018_

## Referee Comment (RC1) · Anonymous Referee #1 · 25 Mar 2018

A really interesting paper that describes modeling efforts to describe oceanic Hg cycling, making use of recent data as well as isotopes. There have been a couple of these in recent years, each using different models and emphasizing different aspects of data as important constraints. This is the first to incorporate isotopic data, which is still very sparse, but has the potential to provide unique boundary conditions that might be very powerful.

The overall approach is sound and the experiments conducted with the model were appropriate and interesting. However, I take issue with some of the formulations and the source functions used to drive the model. I don't have specific fixes to recommend, but I think there's enough data pointing in other directions that I was disappointed the authors didn't game out other scenarios. The authors argued that the engine behind their

experiments, the HAMOCC model of the late Ernst Maier-Reimer and colleagues, is fast and coarse enough to make it particularly useful for trying all sorts of scenarios...so this added to my disappointment.

The authors used a particularly low particle partition coefficient (Kd)...even lower than that used by other modelers and whose values were recently suggested to be too low to begin with. As the authors very briefly suggest, this must imply that their supplement of particles onto HAMOCC is not quite right, but then leave that point. In order to get the percent of particulate Hg right, and using a very low Kd value, their model must be drastically overestimating the amount of particulate matter in the ocean. Perhaps this is a situation of "two wrongs making a right" but the authors should have spent more time examining and explaining this situation and whether it creates downstream problems.

• Formulation criticisms...not necessarily to be changed, but perhaps to be discussed: 1) In Figure 1 and in the text, the authors argue that they will mostly adopt the reaction scheme developed by Semeniuk and Dastoor. They later make some changes (loss of demethylation to Hg(II)) to reconcile isotopic trends. That modification flies in the face of data, both spiking experiment results (Monperrus, Lehnherr, Perrot) and distributional data. This last point is perhaps best illustrated by noting that in Figure 1, using the relative reaction rate constants listed would imply steady state (in the dark) MeHg/Hg(II) ratios of 0.2 (which are too high compared to real data); Hg0/Hg(II) ratios of 0.01 (too low); DMHg/Hg(II) ratios of 0.3 (too high). So, while I appreciate the difficulty of developing a reaction scheme, the authors seem to have adopted an approach (from S&D) that probably was not quite right to begin with and felt compelled to modify it to reconcile with isotopic data. I would have liked to have seen them play with the reaction scheme in various ways to see if they could get it to fit real data both in terms of totals, species and isotopes. This feels like a missed opportunity. 2) The total amount of Hg in the ocean seems much too high in the Present Day scenarios (e.g., Figure 2). Both the Atl and Pac profiles have maxima that are just not seen in the ocean. Deep waters are close to right, but surface waters are much too high, and the subsurface

max is also too high by perhaps a factor of two or more. This is probably the result of using the Streets emissions, which some are arguing does not represent how much Hg is actually moving around the planet (probably a lot is sequestered in soils). But, the authors do not comment on this inconsistency.

• Minor points In several locations, in-text citations are inconsistent in format. Sometimes single name for a multi-author article, sometimes not.

Page 1 Line 11, perhaps use "a" instead of "an" in front of "Hg," since someone reading the text would say "a mercury maximum" instead of "an 'h' 'g' maximum."

Line 21, "it is the only heavy metal known to magnify its toxicity by bio-accumulating up the food chain..." not true...most heavy metals undergo some level of bio-accumulation, and bio-magnification (the latter is the term I think the authors mean to use). It may be the most dramatic example, but it's not the only example.

Line 25, the references cited for the various pollution sources of Hg are mostly derivative references...Streets 2017 is probably all that's needed there.

Line 27, citing Streets for the factor of 3-5 increase is not accurate. They do not make a claim about that in that particular paper. That team also mostly favors a much larger degree of perturbation, as exemplified in Amos 2013, with even larger values implied with the enhanced emissions suggested by Horowitz and in the Streets reference.

Page 2 Line 18, extra left parenthesis before "Kwon"

Line 20, "six at high abundance (<10%)"...change "<" to ">".

Line 32, after all that info in the previous few sentences about isotopes, a reference is probably needed...Blum of some sort?

Page 3 Line 2, hyphen missing in "Maierreimer"

Line 16, extra left parenthesis before "but". Also, the authors say that they will describe differences in their model compared to Semeniuk and Dastoor in the Results section.

Wouldn't it be better to talk about that here in the Methods?

Line 20, "the rate constant for MMHg production from Hg(2+) is proportional to the rate of POC degradation..." This statement is not support by data...though it may be a reasonable place to start in terms of modeling. Some of this comes from Sunderland North Pacific data, but that trend is not universal, and really only speaks to steady MeHg concentrations, not the rate constants. If the sentence is meant to talk about how the methylation is set to be, then the sentence should be "is taken to be proportional to the rate of POC degradation..." and maybe add some caveats in there about "who really knows..." or something like that.

Line 21, "Other Hg transformation reactions are provoked by light, and only take place near the surface ocean..." This too is not supported by data. Furthermore, one of the other models out there by Zhang et al found that subsurface interconversion between Hg(II) and Hg(0) was key in getting the right Hg distributions in the ocean. Thus, this model is ignoring chemistry that other groups have already suggested happens and maybe critical to overall Hg cycling.

Line 29, "the rates of gas evasion...are taken to be proportional to the concentration of the species [should add "ocean concentrations," here], on the assumption that atmospheric concentrations are negligible." I understand what the authors are saying here, but the word "assumption" seems a little weird. We know that gas-exchange of Hg between ocean and air is under water-side control and that atmospheric concentrations are not so much negligible but constant, so that it can be easily folded into the flux calculations. There are large seasonal changes in surface Hg0 concentrations, however, and mid- to high latitude waters in winter time are much closer to equilibrium with the atmosphere than during summer, so if the calculations are gradient based and are assuming concentration in the air is zero, then the model is currently overestimate evasion.

Line 29, "driven by deposition...uniform rate around the world." This might be fine at

this level of complexity, but it should be acknowledged that this is a clear distinction from reality. Deposition is known to have strong latitudinal gradients in concentration and in flux...larger fluxes at lower latitudes than at high latitudes.

Page 4 Line 10, "a factor of 3.96 scaling..." Not sure where this comes from...and wouldn't there be two values (one for 199 and one for 202)?

Line 16, the matrix could be set up more for the uninitiated...sentence leading into it is "the equations are..." but we see matrix instead of equations. Perhaps say explicitly what the equations are before rendering them in matrix notation...or skip the matrix notation altogether and show the equations (folks who know matrix notation would be able to turn the equations into your matrix without being shown it).

Matrix, no kM2 or kD2? Demethylation to Hg(II) is probably the dominant demethylation pathway...as shown by spiking experiments. The D2 pathway is speculative...we don't actually know much about where D comes from or goes to. Much later in the text, the authors argue that the isotope patterns don't work if there's a kM2 term. That argument should at least be alluded to here to avoid confusion. But, the larger problem is that we know that pathway exists and is pretty important. So, there must be some other thing going on to get the isotope patterns to work out right. There are other problems with the Semeniuk and Dastoor rate constants, apparently, but this pathway seems to need to be removed to make the isotopes work. Exploring this more deeply seems like huge missed opportunity of this paper, because we know this pathways exists...so what would the authors suggest is missing from the reaction scheme that would make the speciation and isotopes work out?

Line 20, strictly speaking, Hg does not "evaporate" from the ocean, it "evades." Evaporation is liquid to gas conversion, not solute to gas.

Line 20, kb is not defined.

Page 5 Matrix, again, no M2 term.

[Figure]

Line 23, HAMOCC is not capitalized.

Line 27, "sinking velocity...is much slower than the actual inferred..." This is backed up by geochemical data like that cited by Anderson. . .this sentence makes it sound like the references ignored the trap data and the discrepancy. Traps are biased to heavily undersample slow sinking material (there's a large body of literature addressing this). The material collected in traps is perhaps representative of sinking material in general, but inferred rates are not necessarily representative.

Page 7 Line 6, "Motivated by reconstructed history...(Streets)...we subject our model to a 4.5x increase..." 4.5x is not really a Streets advocated value. He and his collaborators think it is much larger, perhaps 7x. Others think it is smaller, closer to 3x (Engstrom and others).

Line 19, "a peculiarity....net sea surface fluxes of Hg are always balanced" Unclear to me why this was being brought up...do the authors think this isn't the way nature is? I guess the word "peculiarity" here is confusing me.

Line 22-24, "because Hg concentrations in the top box...will underestimate the Hg surface concentrations..." Not sure I follow this. Why couldn't upwelling terms be included in mass balance for surface Hg concentrations? This was just a choice made? Why that particular streamlining? Do the authors have a sense of how big a difference that might make?

Page 8 Line 11, too many parentheses around Schartup reference.

Line 12, "can be shut down by complexation with dissolved sulfide" this is not true. Especially see work of Heimburger and colleagues. . .methylation in euxinic water does happen. The trend/effect argued for here is seen in sediments, but is apparently not true in water. This is consistent with culture experiments that suggest Hg complexed with sulfide or thiols is fairly bioavailable (Schaefer, Hsu-Kim). Not sure why the sediment data are different. The Lamborg Black Sea data is apparently wrong on Hg

speciation. Also, in the previous sentence, Chakraborty's data says the opposite, production is lower in low O2 environments, as evidenced by %MMHg...a similar story to Sunderland's Pacific data. Low O2 environments are not necessarily strong MeHg production locations on a specific rate basis...there's more MeHg there because there's more total Hg there. % MeHg is not always enhanced in low O2 environments.

Line 21...title "particle sinking vs. the overturning circulation" Why is this section setup as a competition between these two forces? One doesn't steal Hg from the other...they both act to move Hg down into the interior, one is just isopycnal and the other is diapycnal.

Page 9 Line 2, "there are some regional variations in Hg, but they are not systematic, as compared to the clear Pac-Atl differences exhibited by nutrient-type elements..." Not true...the Lamborg 2014 paper specifically made the counter argument. The distribution of Hg is the nutrient like distribution with a transient anthropogenic signal superimposed.

Section 3.3, this first paragraph is written in a way that's a bit hard to follow. For example "some fractionation effect that we impose in the model pulls the isotopic composition of either Hg0 or DMHg away from these values, and the other has to go the opposite way to compensate..." What is the "some fractionation effect"? And what is the "other" that "has to go the opposite way..." I think I know what they are talking about, but this is a bit to imprecisely written to be clear. Also, why can't particle sinking remove isotope signal from the surface as well? Wouldn't the isotope balance be between atmo inputs, and the combined effect of evasion and particle sinking (source vs all sinks)?

Page 10 Line 1, "D199" should be capital delta 199, I think. Also, "it has been observed that there is a large difference in the D199 values of MeHg vs Hg(II) in ocean surface water. Do we know the D199 of Hg(II) in ocean water? I guess fish could be used as a proxy for the D199 of MeHg in the water (is there really no MIF during uptake and biomag?), but do we have any data for the Hg(II) pool? I don't think the Blum 2013

paper has any in it.

Line 2, too many parentheses.

Figure 2, would be nice to see a separate panel of Present/Preanth as a function of depth, to see how deep the anth signal goes.

Figure 4, what are the values of Kd for the lines other than the red one?

Figure 5, does "Total Hg" mean in the entire ocean? Is this the pre-anth ocean, or after some amount of human emissions? The value, between about 1.5 and 2.3 Gmoles is reasonable on the low end, but too high on the high end compared to data.

Figure 6 is mentioned only briefly, and it's difficult to understand the point being made. Are we meant to expect a certain cross-over point? What is being tested in the figure?

Figure 7. Data don't really support a plume of particularly high MeHg in the eastern equatorial Pacific...not remarked upon in the text.

Figure 8, unit is meant to be pmole/L. I realize changing the model will give big differences in the concentrations, but the No Advection scenario is so high that this probably is another reason to be suspicious of the Streets inventory.

Figure 9, similar to Figure 6, no real guidance is given on how to interpret these results, and no direct comparison to data is made either.

Figure 10, no unit on y axis.

Figures 11-15, Figure 11 is discussed in some length, but the others are not discussed extensively. This is disappointing as the incorporation of isotopic data is what is unique about this model. I realize data are scarce, but what would be extremely useful is if this model made some specific predictions about isotopic values of species or locations that could then be the target of measurements.

―――――――――――――――

---

## Referee Comment (RC2) · Anonymous Referee #2 · 9 Apr 2018

Archer and Blum present model simulations for marine Hg speciation and isotopes using the fast, coarse resolution Hamocc model. The prospect in doing so is exciting, and commendable, but as I read through the MS the long list of model limitations (no real air-sea exchange, forced steady state, no upwelling etc.), the total lack of discussion of Hg speciation observations, and the handful of relevant Hg isotope observations. . . .it all leaves me wondering how useful this MS will be to the community. The MS reads like a special issue rush job; did J Blum even read the final draft? Hg isotope notation is often incorrect; Hg isotope phrasing is awkward if not incorrect; units are lacking throughout; numerous statements are unreferenced; in fact most of the relevant marine Hg literature and paradigms are not discussed. We don't even know who or what generates dimethyl-Hg; so it's fine to speculate on this, but it needs to be properly

argumented or other scientists risk considering all assumptions for truths. Despite all that, I enjoyed reading the sensitivity analysis and the insights on how transformation and isotope fractionation processes balance out. With all its shortcomings, this study will stimulate others and help the field move forward.

Detailed comments: P1, L11. Write . . .atmospheric Hg deposition rate. . .' P1, L12. "Hg particle transport has only a relatively small impact on anthropogenic Hg uptake" Suggest . . ..Hg uptake by biota. . .. Also, this conclusion seems to contradict the previous phrase that states that Hg particle transport is essential to reproduce the nutrient type marine Hg profile. P1, L13 check . . .for long after. . .. (instead of longer) P1, L16. Final phrase of abstract is not informative. Try to write on what we can learn from the Hg isotope model version. P1, L21. "only heavy metal known to magnify its toxicity by bio-accumulating" Seems like a strange expression; not sure toxicity can be magnified. Suggest to revise.. P1, L27. Suggest to also cite Lamborg et al., 2014 Nature, who used a different method to assess the same enrichment. P1, L29. Hg is a soft acid, but not a soft atom. Revise. P2, L4. Suggest to add Obrist et al., 2017 Nature reference to the permafrost statement. P2, L18 Opening phrase of Hg isotope section: "Stable isotopes provide a powerful tool for determining the origins ((Kwon et al., 2014;Li et al., 2014;Sherman et al., 2015;Sherman et al., 2013;Balogh et al., 2015;Demers et al., 2015;Donovan et al., 2014) and transformations (Kwon et al., 2013;Kwon et al., 2014) of Hg in the natural environment."

Might it be that other groups than the Blum group also published interesting papers on Hg isotopes?

P2, L25-32 The NVE and even-MIF statements need proper referencing: Schauble 2007, GCA; Estrade 2009 GCA, Zheng and Hintelmann 2010 JPChem etc..

P2, L31 "Even-MIF has only been observed in atmospheric samples"

Even-MIF has also been observed in reservoirs that receive atmospheric deposition, including organic soils (Zheng et al., GBC 2016) and coastal sea water (Strok et al.,

2015, CRG)

P3, L17 Marine Hg(0) is not metallic, because it is a mono-atomic dissolved gas.

P3, L18. "The rates of the dark reactions are correlated to each other and to the overall rate of metabolic activity in the (Semeniuk and Dastoor, 2017) model and ours, with the scaling factors as shown in Figure 1."

Suggest to start the phrase with 'We assume. . .'. Figure 1 caption mentions rate constants; here the text mentions scaling factors. This is unclear; add units for rate constants (1/h, 1/s ?) to caption and text.

P3, L28. "The rates of gas evasion of Hg(0) and DMHg are taken to be proportional to the concentrations of the species, on the assumption that atmospheric concentrations are negligible. The Hg system in the surface ocean is driven by deposition influx of Hg(2+), which is applied at a uniform rate around the world."

Maybe mention that in Hamocc the atmosphere is a boundary condition, which is the reason for these assumption I suppose. That said, these assumption are known to be incorrect.

Section 2.2.1 Please provide units for all parameters discussed: Hg, POC concentrations, rate constants, scavenging constants, sinking velocity, rainfall Hg deposition etc..

P6, L1-5 Plz provide units of Kd.

P6, L10. Mention explicitly what the sedimentation flux of Semeniuk 2017 was. . ..to help put Figure 5 in context. It would also be helpful to have now what the observed (or accepted estimate) flux is (see remarks below).

Figure 4 and caption. What are the colors?

Figure 2,4,5 and associated main text. It would be useful if benchmark observations were shown and discussed.

For ex. the Hg literature over the past two decades states a marine global sedimentation flux of 1 Mmol/y (ex. Mason et al., 2012 review), which in Fig5 corresponds to a sinking velocity of 800 m/day, and not the 400 m/day that is preferred by the authors.

Another example from Figure 2. The simulated oligotrophic atlantic has surface enrichment in THg by 5pM. Observations in the same area (Bowman et al. 2014) show surface depletion at 0.5 pM.

P8, L7. "By construction in the model, and apparently to a large extent in the real ocean, the reactions that produce and consume MMHg in the dark are biologically mediated and selfbalancing, resulting in a nearly uniform proportion of MMHg of total Hg in the deep ocean."

This is very well phrased….it just needs some references to observations.

P8, L15 "The top panel shows the relative change in MMHg concentration at 250 meters water depth from enhancing MMHg production in low-oxygen waters (increasing by a factor of 1+exp(-O2/50 micromolar))."

This is an interesting scenario, but what does the 1=exp…parameterization of MMHg reduction correspond to? Where does it come from?

P9, L15. Clarify 'that deposition period' (avoid pronouns)

Figure 10 caption and y-axis need units. A top and bottom panel are discussed in the main text but not Figure.

P9, L28. "A guiding principle in understanding these results is that in the steady state the isotopic degassing flux, a combination of Hg(0) and DMHg degassing fluxes, has to balance isotopically the input by Hg(2+) rain, which is 0.4‰ in d202 and 0.05‰ in ïĄĎ199."

The Hg isotope notation is incorrect here…..did J Blum actually read the paper ïĄŁ The d202Hg and D199Hg numbers for rain are not referenced, and are also

incorrect….they are possibly switched: D199Hg should be 0.4 per mil….. Shouldn't the Hg flux to sediment also figure in the isotope balance?

P10, L4. "When configured in this way, the Hg cycle was unable to fractionate MMHg without Hg(2+) following along, because of this reaction. The isotopic composition 5 of total Hg throughout the entire ocean picked up this fractionation signature.

This way….this signature…..makes it all very unclear. I cannot follow this section.

P10, L9. "This pathway is consistent with the photochemical mechanism for Hg reaction, which involves accepting an electron to form an intermediary Hg(1)."

This needs referencing; what is 'an intermediary Hg(1)'?

Figure 11 caption. Panels J, K, L are not described.

P10, L17. The Hg evasion fractionation factor needs references. Authors reports "Because DMHg is not returned to the Hg pool as quickly in the model as Hg(0), the isotopic deviation in DMHg does not pass to the other pools, which remain near 0 ‰."

Where does evading DMHg go in the model, since there is no atmosphere? In my mind it is not a matter of 'not passing to other pools; rather surface Ocean DMHg is low, therefore the evasion flux is low, and the ensuing isotope imprint of DMHg degassing remains near 0. But I may be wrong..

P10, L22. "Fractionating the reduction step from Hg(2+) to Hg(0) (Figure 11d) has only a slight impact on the isotopic signatures of any of the species, because there is more Hg(0) produced from MMHg than from Hg(2+)."

More Hg(0) produced from MMHg than from Hg(2+) seems odd; I now see in Figure 1 that indeed Hg(2) photoreduction is curiously small. Current thinking (see for ex. Soerensen et al., ES&T 2010), and current marine models have massive, balancing photo-oxidation and photo-reduction fluxes. Why is the photoreduction so small in this model? How to justify this?

Section 3.3.2 The authors need to explain what literature their various fractionation factors are based on.

P11, L3. "while fractionation in MMHg reduction results in Hg(2+) that is isotopically light"

Unclear because the Hg valence in MMHg is also 2. . ...so this can't be reduction. . ...but just demethylation

Figure caption 14. What are a,b,c,d?

---

## Author Comment (AC1) · 27 Apr 2018

We first wish to acknowledge, and apologize for, the rough edges of the manuscript. As guessed by reviewer #2, we did submit this in a rush for a deadline. We are grateful for the many constructive comments and suggestions from both reviewers, and anticipate a much-improved manuscript as a result. In spite of the need for further polish, both reviewers found the results in the manuscript to be interesting and useful, which we are glad to hear.

A substantive suggestion from reviewer #1 will require some new simulations to be added to the study, to explore further the impact of reaction pathways on the isotopic composition of Hg species in the ocean. We did a lot of messing around with the model

that we left undocumented in the paper, and acknowledge that this should be added. In particular, we didn't show in a figure the impact of the original scheme, where MMHg photodegrades to Hg2+ directly without homogenizing with the Hg0 reservoir. We will also expand our discussion of the reaction mechanism, which we think supports the reaction pathway that the isotopes seem to require.

The issue of the Hg binding constant to organic carbon continues to be kind of baffling, but we will try harder to understand and explain the difference between our model and others. A difficulty is that the concentration of POC is a poorly-defined thing in the real ocean, grading continuously as it does between fast-sinking particles that actually carry Hg vertically, to suspended particles that apparently exchange mass with the faster-sinking fraction, to the very refractory dissolved organic carbon. We did not see documentation on the POC concentrations used in previous models, so as time ran short we left it as a probable discrepancy in the POC concentrations in the models, and used in our model the Kd value that reproduces the salient result from the other models of how much Hg is bound versus dissolved (about 5-10%), as well as being internally consistent between the sinking fluxes of Hg and POC. In a resubmission, we will dig deeper to resolve or at least more clearly document this issue.

Another substantive addition that we envision is a series of runs with Hg deposition in different oceans, to map out the sensitivity of the deep ocean concentration to the distribution of surface deposition. The reviewers noted that real ocean deposition is not uniform around the world as we set it up in our model, and that other ocean Hg models use detailed deposition maps from atmospheric models. Given the uncertainty in the Hg deposition field through time, it seems like it would be a useful thing to map out the ocean subsurface Hg distribution sensitivity to surface deposition patterns.

We will also improve the comparison of model results with measured species concentration and isotopic signature data (which may motivate adjustment of the anthropogenic Hg flux as noted by reviewer #1), the citations and discussion of previous literature, and the clarity of the text, in response to the many constructive comments

and suggestions from the reviewers.

---

## Author Response (AR1)

Overview of changes to the model and the manuscript

Both the underlying formulation of the model and its presentation have been extensively reworked in response to the reviewers' comments. We think the paper is much improved and we are grateful to the reviewers for their time and insights.

The kinetics of Hg transformation have been revised, with the largest changes specifically to the pathways for demethylation and the rate of gas exchange. Now the processes of biological degradation of MMHg, producting Hg(II), is distinct in the model from photodegradation, which we treat as a reduction following Bergquist and Blum. In the previous version of the manuscript these were combined, both forming Hg(0), and we reported that changing that combined pathway to produce Hg(II) would alter the $\Delta^{199}$Hg of the Hg$^{tot}$ in the entire ocean. Now that we resolve the distinction between biological and photochemical degradation of MMHg, the issue we reported in the first version has disappeared; either version of the photochemical reaction seems to work in reproducing the isotopic signature observed in surface waters.

Reviewer #1 was disappointed that we didn't "game out" the kinetics models more thoroughly, and in response we have done a suite of sensitivity studies of the Hg speciation in the ocean in response to uncertainties in 10 different kinetic rate constants.

We compare our model results with observational data in this version of the manuscript – both concentration data and the $\Delta^{199}$Hg of monomethyl Hg inferred from fish data. Reviewer #1 was correct when pointing out that the concentrations in our deep ocean were too high. It turned out to be caused by an error in the gas exchange coefficient, now fixed, with the high sensitivity of the deep ocean Hg concentration to the gas exchange coefficient documented in the sensitivity section.

An issue that we had previously with the adsorption constant for Hg onto POC has also been resolved (Reviewer #1 caught it, again). The units that are used to report the adsorption of Hg onto particles are somewhat strange, requiring POC to be in units of kg/L.

We have added the $\Delta^{200}$Hg isotopic system to the simulation, which appears to be a recorder of the relative rates of wet and dry Hg deposition, with perhaps the potential to resolve differences between basins in this ratio.

The advantage of the HAMOCC code is that it is fast enough to do many studies. We believe the paper now makes better use of this strength, by presenting many sensitivity runs, to particle sinking rate, the adsorption constant, kinetic rate parameters, and isotopic fractionation mechanisms.

Responses to specific reviewer comments – Reviewer #1

Page 3: on subsurface interconversion between Hg(II) and Hg(0); this is included in the model, as it also was in the previous version. There was a confusing sentence in the manuscript, now gone, which seemed to imply to the reviewer that there was no interconversion between Hg(II) and Hg(0) in subsurface waters, but what the sentence was trying to say was that photochemical reactions only happen in surface waters.

Page 7: The section on method limitations has been clarified.

Page 8: The section from the previous manuscript about MMHg production in low-O2 conditions has been removed, replaced with the kinetics sensitivity studies. O2 in HAMOCC is problematic in that it goes slightly negative in the equatorial Pacific, and our previous formulation of just increasing MMHg production rates when O2 goes low was too simple. Rather than show the MMHg sensitivity to an unrealistic kinetics change, better to show profiles of all Hg species as they respond to many different kinetics changes.

Page 9 Line 2, "there are some regional variations in Hg, but they are not systematic, as compared to the clear Pac-Atl differences exhibited by nutrient-type elements..." Not true...the Lamborg 2014 paper specifically made the counter argument. The distribution of Hg is the nutrient like distribution with a transient anthropogenic signal superimposed.

That sentence referred to a scenario with only advection, no particles. In Figure 2 one can see a nutrient-type distribution of deep Hg in the full model, as a function of the sinking rate of particles. Looking at the data, however, there are a lot of high values in the North Atlantic, so the nutrient-type distribution is not totally clear.

Section 3.3, this first paragraph is written in a way that's a bit hard to follow.

This section is one of several in the paper that has been extensively rewritten; we hope it is clearer now.

Page 10: added reference to Medina et al paper submitted to *Global Biogeochemical Cycles* that includes measurements of $\Delta^{199}$Hg of Hg(II) on particles. There is also a new figure comparing the model results with those observations.

Figure 6 and 9: A sentence has been added explaining the point of figure 6, and the previous figure 9 has been subsumed into figure 3, but the point of it is also explained in the text.

Figure 11-15 and isotopic data. We have added a comparison with MMHg $\Delta^{199}$Hg measurements from fish, and we show profiles, maps, and summaries, in the hopes of showing patterns that people could look for in new measurements.

Reviewer #2

Comments and suggestions by reviewer #2 were extremely useful and helpful, and were pretty much entirely adopted in the text.

Figure 6 now has data and other model results for comparison.

[revised manuscript text omitted]

david archer 8/21/2018 8:55 AM

david archer 8/21/2018 8:55 AM

david archer 8/21/2018 8:55 AM

Joel Blum 8/4/2018 1:18 PM
Comment [2]: since we don't know anything about DMHg isotopes we could assume it all degrades to MMHg?? OK No Problem.

david archer 8/2/2018 1:28 PM
Comment [3]: But since it's a gas we know it will evade, and that alternative pathway has interesting impacts on the isotopes, and anyway everything would have to be re-done to take it out. Which I could do easily, it's all automated, if there were a strong reason to.

david archer 8/21/2018 8:55 AM

david archer 8/21/2018 8:55 AM

Joel Blum 8/4/2018 1:22 PM
Comment [4]: I think this is OK, but how [20]

david archer 8/21/2018 8:55 AM

Joel Blum 5/15/2018 2:24 PM
Comment [5]: why not carry the delta va [22]

Joel Blum 7/31/2018 1:57 PM
Comment [6]: I don't understand howyo [23]

david archer 8/21/2018 8:55 AM

Joel Blum 5/15/2018 2:39 PM
Comment [7]: as we discussed could add [25]

david archer 8/21/2018 8:55 AM

Joel Blum 7/31/2018 2:12 PM
Comment [8]: change Hg(2+) in Figure to Hg(0)

david archer 8/21/2018 8:55 AM
$$\begin{bmatrix} -k_{20}-k_{2M}-k_{2D}-S & k_{02} \\ k_{20} & -k_{evp} \\ k_{2M} & 0 \\ k_{2D} & 0 \end{bmatrix}$$

david archer 8/21/2018 8:55 AM

$$S = \left(1 - \frac{1}{k_b[POC]+1}\right)\frac{R}{dz}$$

[revised manuscript text omitted]

$$[\text{Hg-P}] / [\text{Hg(II)}] = K_d\,[\text{POC}] \qquad\qquad\qquad (\text{eqn. 1})$$

Semeniuk and Dastoor (2017) and Zhang et al. (Zhang et al., 2014a) used a value of $2 \cdot 10^5$, in units of L kg$^{-1}$ (requiring POC to be in kg L$^{-1}$), but included a factor of 10 correction for the fraction of particulate material in the ocean that is organic carbon ("$f_{oc}$"), resulting in an effective $K_d$ of $2 \cdot 10^6$. Lamborg et al. (2016) derived a value of about $4 \cdot 10^6$, which they claim to be a factor of 20 higher than the value used in the models, but after the $f_{oc}$ correction in the models the values only differ by a factor of 2. The data from Bowman et al. (2015), analysed by Lamborg et al. (2016), showed that about 5% of the Hg(II) in surface waters is bound to sinking particles, similar to the results from the models (Semeniuk and Dastoor, 2017). A map of the bound fraction in surface waters from our model is shown in Figure 6, showing more particulate Hg in high-production (high POC) regions such as the equatorial Pacific. The sensitivity of the model to the value of $K_d$ is shown in Figure 4, with results similar to those for sinking velocity. The calculated lifetime of dissolved Hg in the water column, relative to removal by adsorption to sinking particles, is shown in Figure 7. The figure is intended for comparison with results of other models,

david archer 8/21/2018 8:55 AM

david archer 8/21/2018 8:55 AM

david archer 8/21/2018 8:55 AM

david archer 8/21/2018 8:55 AM

david archer 8/21/2018 8:55 AM

david archer 8/21/2018 8:55 AM

david archer 8/21/2018 8:55 AM

david archer 8/21/2018 8:55 AM
Moved down [1]: (2016) derived a value of about $4\cdot10^6$,

david archer 8/21/2018 8:55 AM

david archer 8/21/2018 8:55 AM
Moved down [2]: (2017)

[revised manuscript text omitted]

david archer 8/21/2018 8:55 AM

david archer 8/21/2018 8:55 AM

david archer 8/21/2018 8:55 AM

david archer 8/21/2018 8:55 AM

Joel Blum 8/4/2018 2:29 PM
**Comment [9]:** can you say something about 198Hg just to satisfy readers curiosity?

david archer 8/21/2018 8:55 AM

david archer 8/21/2018 8:55 AM

david archer 8/21/2018 8:55 AM

david archer 8/21/2018 8:55 AM

david archer 8/21/2018 8:55 AM

david archer 8/21/2018 8:55 AM

david archer 8/21/2018 8:55 AM

david archer 8/21/2018 8:55 AM

david archer 8/21/2018 8:55 AM

david archer 8/21/2018 8:55 AM

david archer 8/21/2018 8:55 AM

Joel Blum 8/6/2018 12:23 PM
**Comment [10]:** ok thanks for writing this. It helps me in seeing behind the curtain.

fractionations are calculated by subtracting the expected mass dependent fractionation, to produce a composite quantity Δ value.  The solver finds the impact of the fractionation mechanisms on the steady-state isotopic signatures in the Hg system: the expression of the isotope effects within the kinetic ocean Hg cycle.

**2.4 The Anthropogenic Perturbation**

Human activity has resulted in significantly increased Hg emission to the global biosphere since about 1850 (Streets et al., 2011;Streets et al., 2017;Amos et al., 2013;Horowitz et al., 2014), which has lead to an increase in Hg deposition to the ocean.  Because of the tendency for Hg to recycle in the environment, the relationship between emissions and deposition is not simple and immediate, but rather reflects the entire cumulative emission and re-emission of Hg.  Guided by a reconstructed history of atmospheric Hg through time (Streets et al., 2017), we subject our model to a 4 times increase in Hg deposition, following an initial spin-up equilibration period of 10,000 years.  The beginning of the anthropogenic period corresponds to approximately the year 1850.  We show natural steady-state results from model year 1850, which are useful for understanding how the ocean Hg cycle works, and contemporary results from model year 2010, for comparison with field measurements.  Anthropogenically enhanced deposition is continued at a constant rate until the year 2100, after which we follow two scenarios: an abrupt and unrealistic return to natural Hg deposition fluxes, useful to determine the time constant of the oceanic recovery, and a "hangover" scenario in which an abrupt cessation of human Hg emissions triggers a gradual slowdown of enhanced deposition, over an ocean overturning time scale of 1,000 years.

**2.5 Method Limitations**

The steady state assumption in the Hg solvers limits the ability of Hg-HAMOCC to explore detailed shallow-water interactions of turbulence, ventilation, and photochemistry, and the physics of the tracer advection code preclude exploration of processes on short time scales, such as the seasonal cycle near the surface.  The model allows us to explore the interaction of the Hg chemistry and particle adsorption with the ocean circulation on long time scales.

A peculiarity of the surface ocean solver is that fluxes of Hg across the sea surface are always locally balanced, by construction, neglecting the impact of any upwelling Hg driving sea surface Hg concentrations and evasion rates to higher values.  Similarly to the treatment of $O_2$ gas in HAMOCC (Maier-Reimer and Bacastow, 1990), the Hg concentrations in the surface box (50 m) are maintained at atmospheric saturation through the iterations in the advection scheme.  The concentrations in the box below that (to 125 m) are comprised of 25% saturation while the other 75% is driven by subsurface advection.  Because Hg concentrations in the top box are determined by a balance of fluxes with the atmosphere, in places where surface divergence brings up Hg from below, the advective upwelling source is missed by Hg-HAMOCC, which will underestimate the Hg surface concentrations and degassing rates somewhat.  To use the model to simulate a transient uptake of Hg by the ocean in response to a change in the surface rain rate, we can track the change in global ocean inventory of Hg with time, but the fluxes determined by the solver at the air-sea interface will balance to zero, locally and at all times, defiant

david archer 8/21/2018 8:55 AM

david archer 8/21/2018 8:55 AM

david archer 8/21/2018 8:55 AM

david archer 8/21/2018 8:55 AM

david archer 8/21/2018 8:55 AM

david archer 8/21/2018 8:55 AM

Joel Blum 5/16/2018 9:54 AM
Comment [11]: I see the inflection in lake seds at 1850, which is typically called the tart of the industrial revolution

david archer 8/21/2018 8:55 AM

david archer 8/21/2018 8:55 AM

david archer 8/21/2018 8:55 AM

david archer 8/21/2018 8:55 AM

david archer 8/21/2018 8:55 AM

david archer 8/21/2018 8:55 AM

david archer 8/21/2018 8:55 AM

david archer 8/21/2018 8:55 AM

david archer 8/21/2018 8:55 AM

david archer 8/21/2018 8:55 AM

Joel Blum 8/4/2018 3:25 PM
Comment [12]: I changes but to and—pl ... [35]

david archer 8/21/2018 8:55 AM

david archer 8/21/2018 8:55 AM

david archer 8/21/2018 8:55 AM

david archer 8/21/2018 8:55 AM

of the net fluxes that are filling the deep ocean with Hg.  The top box of the model (50 m) serves as a sort of boundary condition for Hg.

**3 Results**

**3.1 Particle Sinking versus the Overturning Circulation**

There are two competing mechanisms for Hg invasion into the deep ocean: advection by the overturning circulation and the flux of Hg adsorbed on sinking particles.  We use our model to explore the interaction of these pathways.  There are two end-member cases to consider; one with particles dominating the distribution and transport of Hg, and the other with circulation dominating.  The particle-flux dominating end member condition can be achieved in Hg-HAMOCC by disabling the advection of the Hg tracers (Figure 8, orange line).  In the steady state, in order to achieve Hg concentrations that are not changing through time, the vertical flux of Hg through the water column must be the same at all depth levels.  The flux of sinking POC decreases with depth in the ocean, because it gets eaten.  The abundant POC sinking flux in the surface ocean carries the same Hg sinking flux as the rarefied POC sinking flux in the deep sea.

This means that in the steady state, the POC in the deep sea has to carry more Hg than it would in the surface ocean.  The adsorbed Hg is linearly related to the dissolved Hg by the adsorption equation (1).  Rearranging (1) gives:

$$[Hg\text{-}P] = [Hg(II)] \, Kd \, [POC] \hspace{3cm} (eqn. 2)$$

If we take the sinking Hg-P flux to be proportional to [Hg-P] (assuming a uniform sinking velocity), then a decrease in the flux of POC (proportional to [POC] for the same reason), requires a higher dissolved [Hg(II)].  The result is that, in the steady state, Hg concentrations rise with depth in the ocean, to compensate for the decrease in sinking POC flux.  A smaller POC sinking flux will have to carry a higher Hg concentration in order to sustain the required depth-uniform Hg flux, and the higher adsorbed Hg concentration requires a higher Hg concentration in the water column.

The other end-member case comes much closer to the observed distribution of Hg in the deep ocean.  When circulation dominates, and particle transport of Hg is disabled, the Hg concentrations maintained in the surface ocean (by balancing evasion against deposition) are imposed on the deep ocean, resulting in a nearly uniform distribution of Hg throughout the ocean (Figure 8, blue line).  There are some regional variations in Hg in this scenario, but they are not systematic, as compared to the clear Pacific-Atlantic differences exhibited by nutrient-type elements (concentrated in the Pacific) versus by strongly scavenged elements like Al (concentrated in the Atlantic, where deposition is more intense).

The balance between advection versus sinking particles affects the uptake of anthropogenic Hg by the ocean.  Profiles of total Hg changes from pre-anthropogenic to present-day, after 130 years of enhanced Hg deposition, are shown in Figure 3.  If particles are neglected or sink so slowly as to be negligible in the Hg cycle, there is a sharp surface spike in Hg concentrations in the model simulation of the present-day (2010), due to increased deposition.  An increasing importance of particle transport tends to moderate a surface ocean spike, while transferring much of the anthropogenic Hg load to a

david archer 8/21/2018 8:55 AM

david archer 8/21/2018 8:55 AM

david archer 8/21/2018 8:55 AM

david archer 8/21/2018 8:55 AM

david archer 8/21/2018 8:55 AM

david archer 8/21/2018 8:55 AM

david archer 8/21/2018 8:55 AM

Joel Blum 5/16/2018 10:21 AM
**Comment [13]:** shouldn't this be 2018-1880 = 138 years?

david archer 8/2/2018 1:43 PM
**Comment [14]:** I've seen 2010 as present day in Hg modelling, maybe because it sort of represents when the measurements were made.  We could do 2018 or whever.

david archer 8/21/2018 8:55 AM

david archer 8/21/2018 8:55 AM

david archer 8/21/2018 8:55 AM

Joel Blum 7/31/2018 4:37 PM
**Comment [15]:** wording needs attention here

david archer 8/21/2018 8:55 AM

david archer 8/21/2018 8:55 AM

david archer 8/21/2018 8:55 AM

[revised manuscript text omitted]

Joel Blum 7/31/2018 1:50 PM
Comment [17]: use either Hg²⁺ or Hg(II), but not Hg(II)

david archer 8/21/2018 8:55 AM

david archer 8/21/2018 8:55 AM
Deleted: It has been observed that there is a large difference in the D199 values of MMHg versus Hg(2+) in ocean surface waters (MMHG results from fish: (Blum et al., 2013)). We were forced to modify the web of reactions and kinetics proposed by (Semeniuk and Dastoor, 2017), in particular the formulation of the interreaction between Hg(2+) and MMHg. In (Semeniuk and Dastoor, 2017), MMHg is demethylated to Hg(2+) directly. When configured in this way, the Hg cycle was unable to fractionate MMHg without Hg(2+) following along, because of this reaction. The isotopic composition of total Hg throughout the entire ocean picked up this fractionation signature. A solution is to alter the reaction web so that demethylation produces an intermediary Hg(0), which homogenizes with the rest of the Hg(0) pool before fueling a necessary next step to Hg(2+), which takes place at a faster rate constant than in (Semeniuk and Dastoor, 2017), to compensate for the greater overall traffic of Hg(0) going to Hg(2+). This pathway is consistent with the photochemical mechanism for Hg reaction, which involves accepting an electron to form an intermediary Hg(1). [40]

david archer 8/21/2018 8:55 AM

Joel Blum 5/16/2018 11:08 AM
Comment [18]: this sentence needs clarification/reorking

david archer 8/21/2018 8:55 AM

david archer 8/21/2018 8:55 AM

david archer 8/21/2018 8:55 AM

[revised manuscript text omitted]

Joel Blum 7/31/2018 3:21 PM
Comment [23]: d202 needs to be d202Hg and D199 needs to be D199Hg and so-on.

Joel Blum 8/4/2018 11:52 AM
Comment [24]: On the fugure you need to add Hg where=ever you have delta values.

david archer 8/21/2018 8:55 AM
Moved up [6]: Figure 11.

david archer 8/21/2018 8:55 AM
Moved (insertion) [6]

david archer 8/21/2018 8:55 AM
Deleted: Schematic of the expression of isotopic fractions on the isotopic signatures of the Hg species in the model. Fractionations are shown in red, expressed as per mille between 202 and 198. Resulting average $\delta^{202}$ values for each species are in black. A) Mass dependent fractionation applied to Hg(0) evasion (Wiederhold et al., 2010). B) applied to DM Hg degassing (assuming the same fractionation as for Hg(0) evasion). C) Both gas evasion fractionations applied simultaneously. The results in C are the sum of those in A and B. D) Fractionation is applied in the reduction of Hg(2+) to form Hg(0) (Kritee et al., 2007); E) in the methylation of Hg(2+) to form MMHg (Kritee et al., 2009); F) in demethylation / reduction of MMHg to form Hg(0). G) shows the impact of fractionation in particle adsorption. H) shows photochemical demethylation $\Delta^{199}$ MIF mass independent fractionation results, while (I) is the effect on $\delta^{202}$ from that process , using the ratio of $\Delta^{199}$ to $\delta^{202}$ that results and (i) Hg(2+) reduction. ... [53]

[Figure]

Figure 11. Continued.

[Figure]

Figure 12. Profiles of $\delta^{202}$Hg(II) and $\Delta^{199}$Hg(II) for different fractionation scenarios, global mean, and for the locations in Figure 4.

david archer 8/21/2018 8:55 AM

[Figure]

[Figure]

**Figure 12.** Continued.

[Figure]

Figure 13a.  Maps of $\delta^{202}$Hg(II) at the sea surface (left) and at 3 km depth (right) for different fractionation scenarios.

david archer 8/21/2018 8:55 AM

Surface          3 km

Hg$^0$ evn

DMHg evn

Hg$^{2+}$ redn

MMHg photo red.

Hg$^{2+}$ meth

MMHg bio ox.

Particles

All Frac

[Figure]

[Figure]

**Figure 13b. Maps of Δ$^{199}$Hg(II) at the sea surface (left) and at 3 km depth (right) for different fractionation scenarios.**

david archer 8/21/2018 8:55 AM
**Moved down [7]:** ... [54]

david archer 8/21/2018 8:55 AM

david archer 8/21/2018 8:55 AM

[Figure]

Surface         3 km

Figure 13c.  Maps of $\Delta^{199}$Hg of MMHg at the sea surface (left) and at 3 km depth (right) for different fractionation scenarios.

david archer 8/21/2018 8:55 AM
**Moved down [8]:** Figure 15.

david archer 8/21/2018 8:55 AM

david archer 8/21/2018 8:55 AM

david archer 8/21/2018 8:55 AM

[Figure]

**Figure 14.** **Profiles** of $\Delta^{199}$Hg of MMHg and Hg(II) for **photochemical** fractionation mechanisms. Top row is observations of MMHg $\Delta^{199}$Hg inferred from measurements of fish (Blum et al., 2013, and Δ199Hg for Hg(II) from measurements of Hg on particles [Motta LC, submitted #6704). Second row shows model results with MMHg photoreduction, using isotope fractionation from (Bergquist and Blum, 2007). Third row shows the impact of Hg(II) photo reduction (Kritee et al., 2007). Bottom row shows sum of MMHg and Hg(II) photo reduction mechanisms. Locations for the Eq. Pac. and Olig. Atl. profiles are given in Figure 4.

david archer 8/21/2018 8:55 AM
**Moved (insertion) [7]**

david archer 8/21/2018 8:55 AM

Joel Blum 8/4/2018 12:04 PM
**Comment [25]:** add Hg after 199 in the figure itelf

david archer 8/21/2018 8:55 AM

david archer 8/21/2018 8:55 AM

david archer 8/21/2018 8:55 AM

[Figure]

**Figure 15.** Profiles of the $\Delta^{200}$Hg isotopic composition of Hg(II) for different values of the dry deposition (Hg(0)) flux. The rate of wet deposition is the same for all runs. Locations for the Eq. Pac. and Olig. Atl. profiles are given in Figure 4.

david archer 8/21/2018 8:55 AM
**Moved (insertion) [8]**

[Figure]

Figure 16. The $\Delta^{200}$Hg isotopic composition of the global mean ocean is a function of the ratio of wet and dry deposition fluxes.

[Figure]

Figure 17. Maps of steady-state distribution of $\Delta^{200}$Hg(II) under conditions of a doubling of the Hg(II) deposition flux which is isolated to the Atlantic, Pacific, and Indian oceans. The isotopic signature at 3 km depth reflects differences in surface deposition.

---

## Author Response (AR2)

Overview of changes to the manuscript since the last version

In addition to making all of the corrections and taking the suggestions of the reviewers, we also fixed an error in the code, reran all of the simulations, and replotted most of the figures to be safe.  The only substantive change in the figures however is to Figure 11, which shows isotopic signatures of individual fractionation mechanisms.  The runs and the plots are now configured so that the sum of the individual isotope effects add up to the value that we get from the code when we combine all of the fractionations simultaneously.  (Previously all of the simulations were subjected to fractionated deposition along with their individual fractionation mechanism, which, when the results of the simulations were added together, multiply-counted the deposition fractionation, so they didn't add up to the total. This way seems much clearer.)

**A model of mercury cycling and isotopic fractionation in the ocean**

David E. Archer[1], Joel D. Blum[2]

[1]Department of the Geophysical Sciences, University of Chicago, Chicago, 60637, USA
[2]Department of Earth and Environmental Sciences, University of Michigan, Ann Arbor, Michigan 48109

*Correspondence to*: David E. Archer (d-archer@uchicago.edu)

**Abstract.** Mercury speciation and isotopic fractionation processes have been incorporated into the HAMOCC offline ocean tracer advection code. The model is fast enough to allow a wide exploration of the sensitivity of the Hg cycle in the oceans, and of factors controlling human exposure to monomethyl-Hg through the consumption of fish. Vertical particle transport of Hg appears to play a discernable role in setting present-day Hg distributions, which we surmise by the fact that in simulations without particle transport, the high present-day Hg deposition rate leads to an Hg maximum at the sea surface, rather than a subsurface maximum as observed. Hg particle transport has a relatively small impact on anthropogenic Hg uptake, but it sequesters Hg deeper in the water column, so that excess Hg is retained in the model ocean for a longer period of time after anthropogenic Hg deposition is stopped. Among 10 rate constants in the model, steady state Hg concentrations are most sensitive to reactions that are sources or sinks of Hg(0), the evasion of which to the atmosphere is the dominant sink term in the surface ocean. Isotopic fractionations in the interconversion reactions are most strongly expressed, in the isotopic signatures of dissolved Hg, in reactions that involve the dominant dissolved species, Hg(II), including mass independent fractionation during Hg photoreduction. The $\Delta^{199}$Hg of MMHg in the model, subject to photoreduction fractionation, reproduces the $\Delta^{199}$Hg of fish in the upper 1000 m of the ocean, while the impact of anthropogenic Hg deposition on Hg isotope ratios is essentially negligible.

**1 Background**

The element mercury (Hg) is a powerful neurotoxin (Clarkson and Magos, 2006). When transformed to methyl mercury (MeHg) it is known to amplify its toxicity by bio-accumulating up the food chain. The main human exposure to MeHg is via consumption of high trophic level seafood (Chen et al., 2016;Schartup et al., 2018). Humans have been mining and mobilizing Hg into the Earth surface environment for hundreds of years, as a by-product of coal combustion, and for its use in gold mining, and in products such as electronics and light bulbs (Amos et al., 2013;Driscoll et al., 2013;Krabbenhoft and Sunderland, 2013;Lamborg et al., 2014;Obrist et al., 2018;Streets et al., 2017;Mason et al., 2012). The Hg load in the surface ocean has increased by a factor of 3-5 since the industrial revolution; this represents a massive human impact on the global Hg cycle (Streets et al., 2017).

david archer 9/28/2018 11:15 AM
**Comment [1]:** Rev 1 writes may not be strictly true anymore! Gregoire and Poulain 2016 say they did find a use for Hg by phototrophs…not sure if it's been confirmed.

But maybe it's cleaner to just not discuss metabolic use of Hg at all?

David Archer 9/28/2018 11:14 AM

[revised manuscript text omitted]

**David Archer 10/5/2018 11:09 AM**